# Smart Surrogate Losses for Contextual Stochastic Linear Optimization with Robust Constraints

**Hyungki Im**  **Wyame Benslimane**  **Paul Grigas**
Department of Industrial Engineering and Operations Research
University of California, Berkeley
Berkeley, CA 94720
`{hyungki.im, wyame.benslimane, pgrigas}@berkeley.edu`

## Abstract

We study an extension of contextual stochastic linear optimization (CSLO) that, in contrast to most of the existing literature, involves inequality constraints that depend on uncertain parameters predicted by a machine learning model. To handle the constraint uncertainty, we use contextual uncertainty sets constructed via methods like conformal prediction. Given a contextual uncertainty set method, we introduce the "Smart Predict-then-Optimize with Robust Constraints" (SPO-RC) loss, a feasibility-sensitive adaptation of the SPO loss that measures decision error of predicted objective parameters. We also introduce a convex surrogate, SPO-RC+, and prove Fisher consistency with SPO-RC. To enhance performance, we train on truncated datasets where true constraint parameters lie within the uncertainty sets, and we correct the induced sample selection bias using importance reweighting techniques. Through experiments on fractional knapsack and alloy production problem instances, we demonstrate that SPO-RC+ effectively handles uncertainty in constraints and that combining truncation with importance reweighting can further improve performance.

## 1 Introduction

Large-scale optimization problems are prevalent across various domains, from supply chain management and financial planning to energy systems and transportation. These problems often involve uncertain parameters, which may appear in both the objective function and the constraints. In practice, these unknown parameters are often influenced by contextual data, which can be leveraged to improve decision-making. While integrating contextual data to estimate uncertain parameters in optimization models has gained significant interest in recent years (see surveys by Sadana et al. (2024) and Mandi et al. (2024) and references therein), most of the existing literature focuses on scenarios where these parameters only appear in the objective function. However, in practice, uncertain parameters often arise within the constraints of the optimization problem as well (Wang et al., 2023; Rahimian and Pagnoncelli, 2023). Uncertainty in constraints introduces additional layers of complexity, as the feasibility of solutions with respect to the true parameters is no longer guaranteed.

In this paper, we study an extension of contextual stochastic linear optimization (CSLO) where both the objective function and constraints involve uncertain parameters, each of which can be predicted from contextual feature data. We propose a "Smart Predict-then-Optimize with Robust Constraints" (SPO-RC) approach, a framework that provides a reliable method for handling these uncertainties by using robust constraints. Our approach begins by constructing contextual uncertainty sets for the uncertain constraint parameters using existing techniques such as conformal prediction (Romano et al., 2019; Angelopoulos and Bates, 2021; Fontana et al., 2023). We then formulate a contextual robust optimization problem, which provides a safe approximation to a corresponding contextual chance

constrained problem and serves as our mechanism for decision-making. Within this framework, we introduce new cost metrics and their regret forms – the SPO-RC and SPO-RC+ losses. These are direct extensions of the SPO and SPO+ losses, designed to handle changing feasibility sets that vary with contextual data. The SPO-RC loss measures the decision error induced by following a robust predict-then-optimize approach, and the SPO-RC+ loss serves as a convex surrogate. Although we construct uncertainty sets for the constraint parameters, the true parameters may sometimes lie outside these sets, which can potentially lead to infeasible solutions. To address this issue, we truncate data points where the uncertainty sets do not cover the true parameters, focusing our model's capacity on regions where feasibility is guaranteed. However, the truncation introduces sample selection bias, which we mitigate using importance reweighting techniques, such as Kernel Mean Matching (KMM). Our contributions can be summarized as follows:

1. We propose the SPO-RC approach, which formulates a contextual robust optimization problem for decision-making and enables us to learn the objective cost vector in an integrated manner. We propose a convex surrogate called SPO-RC+, and show that Fisher consistency between the proposed SPO-RC and SPO-RC+ losses holds.

2. We introduce learning with truncation to focus on a region where feasibility is guaranteed, and we propose an importance reweighting technique to mitigate sample selection bias.

3. We present numerical results on the fractional knapsack problem and an alloy production problem. These results demonstrate the relative merits of truncation and importance reweighting, the robustness of our approach, and the potential for combining our methodology with enhanced scalability techniques like solution caching (Mulamba et al., 2021).

## 1.1   Literature review

Contextual stochastic optimization (CSO) involves making decisions under uncertainty using contextual information to improve outcomes. Sadana et al. (2024) offers a comprehensive review of the methodologies developed for CSO, categorizing them into three primary approaches. The first approach, "decision rule optimization", centers on learning a mapping from contextual data to decisions (Liyanage and Shanthikumar, 2005). Research in this area explores a broad spectrum of methods, including linear decision rules (Ban and Rudin, 2019), reproducing kernel Hilbert space (RKHS) based decision rules (Bazier-Matte and Delage, 2020; Bertsimas and Koduri, 2022; Notz and Pibernik, 2022), and non-linear decision rules (Huber et al., 2019; Oroojlooyjadid et al., 2020; Rychener et al., 2023; Chen et al., 2023; Zhang et al., 2023). The second approach, "sequential learning and optimization" (SLO), focuses on learning the conditional distribution of outcomes based on contextual data. Decision-making/optimization methods are built on top of the learning methods, and include residual-based methods, (Deng and Sen, 2022; Kannan et al., 2021, 2025, 2024) and importance weighted methods (Bertsimas and Kallus, 2020). To address challenges such as the "optimizer's curse", robust methodologies have been introduced, including distributionally robust optimization (DRO) and regularization techniques (Nguyen et al., 2024; Bertsimas and Van Parys, 2022; Esteban-Pérez and Morales, 2022; Lin et al., 2022). Lastly, the "integrated learning and optimization" (ILO) approach, also known as decision-focused learning, directly trains predictive models with the goal of improving downstream optimization outcomes. This is achieved by minimizing the expected objective value of the solution induced by the prediction models. However, a key challenge is the computational difficulties of training the model by optimizing a loss function that measures decision error. To overcome this challenges, researchers have introduced implicit differentiation methods (Amos and Kolter, 2017; Agrawal et al., 2019; Blondel et al., 2022; McKenzie et al., 2023; Butler and Kwon, 2023) and differentiable surrogate loss functions (Elmachtoub et al., 2020; Elmachtoub and Grigas, 2022; Loke et al., 2022; Kallus and Mao, 2023; Estes and Richard, 2023). Our work is particularly related to the SPO framework (Elmachtoub and Grigas, 2022), as we extend this framework to handle problems with uncertain constraints. As pointed out by Elmachtoub and Grigas (2022), the SPO framework can also be viewed as an extension of structured prediction (Taskar et al., 2005; Nowozin et al., 2011), where the SPO+ loss is an extension of the structured hinge loss. Parmentier (2022) also uses structured prediction to learn a predictor that maps hard instances of optimization problems arising in operations research to easier combinatorial instances.

Constructing uncertainty sets conditional on contextual information is important to guarantee the feasibility of the downstream optimization task. Recent work on learning conditional uncertainty sets has explored various approaches, including partitioning methods (Chenreddy et al., 2022), the

$k$-nearest ellipsoid method (Ohmori, 2021), differentiable optimization (Wang et al., 2023; Chenreddy and Delage, 2024), and conformal prediction (Johnstone and Cox, 2021; Patel et al., 2024a,b), the latter gaining significant interest due to its distribution-free, model-agnostic characteristics (Sun et al., 2023; Stanton et al., 2023). More recently, end-to-end training methodologies that integrate uncertainty estimation with downstream decision-making objectives have been proposed (Yeh et al., 2024). However, most of these approaches consider uncertainty only in the objective function rather than in the constraints. We introduce a flexible framework that can incorporate multiple methods for constructing uncertainty sets for constraint parameters, with conformal prediction being just one example.

**Notation.** Given a distribution $\mathcal{D}$, we denote the marginal distribution of $\mathbf{x}$ by $\mathcal{D}_{\mathbf{x}}$. Additionally, we use the superscript notation $\mathcal{D}^n$ to represent an i.i.d. dataset of size $n$ sampled from $\mathcal{D}$. We use $\mathbb{1}_m$ to represent the $m$-dimensional one-vector.

## 2 Smart predict-then-optimize with robust constraints

To motivate our methodology, consider the following optimization problem:

$$\min_{\mathbf{w} \in \mathbb{R}^d} \mathbb{E}[\mathbf{c}^\top \mathbf{w} | \mathbf{x}] \quad \text{s.t.} \quad \mathbb{P}(h(\mathbf{w}; \mathbf{a}) \leq 0 | \mathbf{x}) \geq 1 - \alpha, \ \mathbf{w} \in S, \tag{1}$$

where $\mathbf{w} \in \mathbb{R}^d$ represents the decision variables, $\mathbf{c} \in \mathbb{R}^d$ is the uncertain cost parameter, and $\mathbf{a} \in \mathbb{R}^m$ is the uncertain constraint parameter, with $h : \mathbb{R}^d \times \mathbb{R}^m \to \mathbb{R}^l$. The set $S \subseteq \mathbb{R}^d$ is compact and convex. Contextual data $\mathbf{x} \in \mathbb{R}^p$ is used to predict $\mathbf{c}$ and $\mathbf{a}$, with $\mathcal{D}$ denoting the distribution of $(\mathbf{x}, \mathbf{c}, \mathbf{a})$. The objective is risk neutral in $\mathbf{c}^\top \mathbf{w}$, while the constraints must be satisfied with probability at least $1 - \alpha$. Due to the intractability of the chance-constrained problem, we adopt a common technique in the literature to reformulate/approximate problem (1) as a robust optimization problem by constructing an uncertainty set $\mathcal{U}(\mathbf{x})$ such that $\mathbf{a} \in \mathcal{U}(\mathbf{x})$ with probability at least $1 - \alpha$. This is not intended as a precise guarantee–it serves merely as a motivation for the robust formulation that follows. There exist various approaches to construct such an uncertainty set $\mathcal{U}(\cdot)$ from data, including methods inspired by conformal prediction. With a suitable $\mathcal{U}(\cdot)$, we reformulate problem (1) as a robust optimization:

$$\min_{\mathbf{w} \in \mathbb{R}^d} \mathbb{E}[\mathbf{c}|\mathbf{x}]^\top \mathbf{w} \quad \text{s.t.} \quad h(\mathbf{w}; \mathbf{a}) \leq 0 \ \forall \mathbf{a} \in \mathcal{U}(\mathbf{x}), \ \mathbf{w} \in S. \tag{2}$$

Assuming that a tractable robust counterpart exists and that a feasible solution can be found, the solution to problem (2) is feasible with probability at least $1 - \alpha$. The linearity of expectation simplifies the objective from $\mathbb{E}[\mathbf{c}^\top \mathbf{w} | \mathbf{x}]$ to $\mathbb{E}[\mathbf{c}|\mathbf{x}]^\top \mathbf{w}$. Moreover, since the uncertainty set $\mathcal{U}(\mathbf{x})$ depends on $\mathbf{x}$, the feasible set of (2) dynamically changes with $\mathbf{x}$. Note that our methodology, and our proposed SPO-RC and SPO-RC+ losses, build directly on the contextual robust optimization problem (2), a more tractable alternative to problem (1).

**Remark 2.1.** *For a precise guarantee that the conditional chance constraint $\mathbb{P}(h(\mathbf{w}; \mathbf{a}) \leq 0 | \mathbf{x}) \geq 1 - \alpha$ holds for each $\mathbf{x}$, one would require that $\mathbb{P}(\mathbf{a} \in \mathcal{U}(\mathbf{x}) \mid \mathbf{x}) \geq 1 - \alpha$. While achieving this stricter conditional coverage may limit the range of techniques available for constructing $\mathcal{U}(\cdot)$, the approach we present is compatible with both marginal and conditional coverage assumptions.*

### 2.1 Cost metrics

We focus on developing an integrated approach to learn the cost vector by estimating the conditional expectation $\mathbb{E}[\mathbf{c}|\mathbf{x}]$ within a robust predict-then-optimize pipeline. To achieve this, we introduce a cost metric that measures the quality of solutions derived from our predictions. We consider a hypothesis class $\mathcal{F}$ of cost vector prediction models, where each model $f : \mathbb{R}^p \to \mathbb{R}^d$ maps a context vector $\mathbf{x}$ to a predicted cost vector $\hat{\mathbf{c}} = f(\mathbf{x})$. Given $\hat{\mathbf{c}}$ and $\hat{\mathcal{U}} = \mathcal{U}(\mathbf{x})$, we define the robust optimization problem $\mathcal{P}(\hat{\mathbf{c}}, \hat{\mathcal{U}})$ as:

$$\mathcal{P}(\hat{\mathbf{c}}, \hat{\mathcal{U}}) : \quad \min_{\mathbf{w} \in \mathbb{R}^d} \hat{\mathbf{c}}^\top \mathbf{w} \quad \text{s.t.} \quad h(\mathbf{w}; \mathbf{a}) \leq 0 \ \forall \mathbf{a} \in \hat{\mathcal{U}}, \ \mathbf{w} \in S.$$

We denote $W^*(\hat{\mathbf{c}}, \hat{\mathcal{U}})$ as the optimal solution set of $\mathcal{P}(\hat{\mathbf{c}}, \hat{\mathcal{U}})$, $w^*(\hat{\mathbf{c}}, \hat{\mathcal{U}})$ as an arbitrary element of $W^*(\hat{\mathbf{c}}, \hat{\mathcal{U}})$. We also define $\mathcal{S} := \mathcal{S}(\mathbf{x})$ as the feasible set for $\mathcal{P}(\hat{\mathbf{c}}, \hat{\mathcal{U}})$. Given the predictions $\hat{\mathbf{c}}, \hat{\mathcal{U}}$

and the realized cost vector $\mathbf{c}$, we introduce the following notation to record the cost of the decision vector $w^*(\hat{\mathbf{c}}, \hat{\mathcal{U}})$ with respect to the realized cost vector $\mathbf{c}$:

$$\text{cost}(\hat{\mathbf{c}}, \mathbf{c}; \hat{\mathcal{U}}) := \mathbf{c}^\top w^*(\hat{\mathbf{c}}, \hat{\mathcal{U}}).$$

The cost metric above is minimized when the predicted cost vector $\hat{\mathbf{c}}$ exactly matches the true cost vector $\mathbf{c}$. However, since computing the cost metric can be intractable due to the non-differentiable and discontinuous nature of the oracle $w^*(\cdot, \hat{\mathcal{U}})$, we introduce the tractable $\text{cost}_+$ metric inspired by the SPO+ loss Elmachtoub and Grigas (2022).

$$\text{cost}_+(\hat{\mathbf{c}}, \mathbf{c}; \hat{\mathcal{U}}) := \max_{\mathbf{w} \in \mathcal{S}} \left\{ (\mathbf{c} - 2\hat{\mathbf{c}})^\top \mathbf{w} \right\} + 2\hat{\mathbf{c}}^\top w^*(\mathbf{c}, \hat{\mathcal{U}})$$

For a detailed derivation of the $\text{cost}_+$ metric, we refer the reader to Elmachtoub and Grigas (2022). The $\text{cost}_+$ metric is convex with respect to $\hat{\mathbf{c}}$, making it learnable using first-order methods.

**Proposition 2.2.** *Given $\mathbf{c}$ and $\hat{\mathcal{U}}$, the $\text{cost}_+$ metric satisfies:*

1. *$\text{cost}(\hat{\mathbf{c}}, \mathbf{c}; \hat{\mathcal{U}}) \leq \text{cost}_+(\hat{\mathbf{c}}, \mathbf{c}; \hat{\mathcal{U}})$ for all $\hat{\mathbf{c}} \in \mathbb{R}^d$,*

2. *$\text{cost}_+(\hat{\mathbf{c}}, \mathbf{c}; \hat{\mathcal{U}})$ is a convex function of $\hat{\mathbf{c}}$ and $2(w^*(\mathbf{c}, \hat{\mathcal{U}}) - w^*(2\hat{\mathbf{c}} - \mathbf{c}, \hat{\mathcal{U}}))$ is a subgradient of $\text{cost}_+$ metric at $\hat{\mathbf{c}}$.*

Given the $\mathcal{U}(\cdot)$ and a dataset $\mathcal{D}_n := \{\mathbf{x}_i, \mathbf{c}_i, \mathbf{a}_i\}_{i=1}^n$, we learn the predictor $\hat{f}$ by solving:

$$\hat{f} \in \underset{f \in \mathcal{F}}{\arg\min} \frac{1}{n} \sum_{i=1}^n \text{cost}_+ \left( f(\mathbf{x}_i), \mathbf{c}_i; \mathcal{U}(\mathbf{x}_i) \right).$$

## 2.2 Fisher consistency

We show that many desirable properties from the standard SPO and SPO+ frameworks carry over to our setting. Specifically, we show that the Fisher consistency of the $\text{cost}(\hat{\mathbf{c}}, \mathbf{c}; \hat{\mathcal{U}})$ and $\text{cost}_+$ metric holds (Elmachtoub and Grigas, 2022).

Assuming full knowledge of the data distribution and no restrictions on the hypothesis class, Fisher consistency implies that minimizing the surrogate loss, the $\text{cost}_+$ metric, also minimizes the true cost metric. This justifies using the $\text{cost}_+$ metric when it is challenging to directly minimize the cost metric. To formalize this, let $f^*_{\text{cost}}$ and $f^*_{\text{cost}_+}$ be the functions that minimize the expected cost and $\text{cost}_+$ metric, respectively:

$$\begin{aligned}
f^*_{\text{cost}} &= \arg\min_f \mathbb{E}_{(\mathbf{x}, \mathbf{c}) \sim \mathcal{D}_{\mathbf{x}, \mathbf{c}}} \left[ \text{cost} \left( f(\mathbf{x}), \mathbf{c}; \mathcal{U}(\mathbf{x}) \right) \right], \\
f^*_{\text{cost}_+} &= \arg\min_f \mathbb{E}_{(\mathbf{x}, \mathbf{c}) \sim \mathcal{D}_{\mathbf{x}, \mathbf{c}}} \left[ \text{cost}_+ \left( f(\mathbf{x}), \mathbf{c}; \mathcal{U}(\mathbf{x}) \right) \right].
\end{aligned} \tag{3}$$

Under the assumption of full knowledge of $\mathcal{D}$ and an unrestricted hypothesis class, we aim to show that:

$$f^*_{\text{cost}} = f^*_{\text{cost}_+} = \mathbb{E}\left[ \mathbf{c} | \mathbf{x} \right] \tag{4}$$

This equality implies that minimizing the cost and $\text{cost}_+$ metric leads to the same optimal predictor, both resulting in $\mathbb{E}\left[ \mathbf{c} | \mathbf{x} \right]$, which is the true optimal solution.

**Assumption 2.3.** *To demonstrate the Fisher consistency of the $\text{cost}$ and $\text{cost}_+$ metrics, we assume the following:*

1. *Almost surely, $W^*(\mathbb{E}[\mathbf{c}|\mathbf{x}], \hat{\mathcal{U}})$ is a singleton for all $\mathbf{x} \in \mathcal{X}$, where $\mathcal{X}$ denotes the domain of the contextual variables.*

2. *The distribution of $\mathbf{c}|\mathbf{x}$ is both centrally symmetric about its mean $\mathbb{E}[\mathbf{c}|\mathbf{x}]$ and continuous over the entire space $\mathbb{R}^d$, for all $\mathbf{x} \in \mathcal{X}$.*

3. *The interior of the feasible region $\mathcal{S}(\mathbf{x})$ is non-empty for all $\mathbf{x} \in \mathcal{X}$.*

**Theorem 2.4.** *Under Assumption 2.3, $f^*_{\text{cost}}$ and $f^*_{\text{cost}_+}$ satisfy (4). Therefore, the $\text{cost}_+$ metric is Fisher consistent with the $\text{cost}$ metric.*

The proof of Theorem 2.4 (and other proofs) is provided in Appendix A. A key aspect of Theorem 2.4 is the assumption that we are working with a well-specified hypothesis class. This ensures that the learned function can capture the true conditional expectation $\mathbb{E}[\mathbf{c}|\mathbf{x}]$ for every $\mathbf{x}$.

## 2.3 SPO-RC and SPO-RC+ loss functions

A natural extension of the cost metric into a loss function, similar to the SPO approach, is to compare the cost incurred by the predicted solution with the optimal cost achievable in hindsight, assuming perfect knowledge of the true parameter $\mathbf{a}$. We formally introduce the loss function called the SPO Robust Constraint (SPO-RC), defined as follows:

$$\ell_{\text{SPO-RC}}(\hat{\mathbf{c}}, \mathbf{c}; \hat{\mathcal{U}}, \mathbf{a}) := \begin{cases} \text{cost}(\hat{\mathbf{c}}, \mathbf{c}; \hat{\mathcal{U}}) - \mathbf{c}^\top w^*(\mathbf{c}, \delta_{\mathbf{a}}), \text{if } \mathbf{a} \in \hat{\mathcal{U}}, \\ \Delta_S(\mathcal{C}), \text{otherwise}, \end{cases}$$

where $\delta_{\mathbf{a}} := \{\tilde{\mathbf{a}} : \tilde{\mathbf{a}} = \mathbf{a}\}$ is a singleton uncertainty set. The function $\Delta_S(\mathcal{C})$ provides an upper bound on the maximum possible difference in objective value over all solutions in $S$ throughout the compact vector space $\mathcal{C}$. It is defined as:

$$\Delta_S(\mathcal{C}) := \sup_{\mathbf{c} \in \mathcal{C}} \left( \max_{\mathbf{w} \in S} \mathbf{c}^\top \mathbf{w} - \min_{\mathbf{w} \in S} \mathbf{c}^\top \mathbf{w} \right).$$

The term $\mathbf{c}^\top w^*(\mathbf{c}, \delta_{\mathbf{a}})$ represents the pure optimal cost, which is achievable in hindsight with perfect knowledge of $\mathbf{a}$. Specifically, when $\mathbf{a} \notin \hat{\mathcal{U}}$, the predicted solution $w^*(\hat{\mathbf{c}}, \hat{\mathcal{U}})$ might outperform the optimal solution in hindsight, resulting in a negative loss. To ensure that the loss remains nonnegative, we set it to the upper bound $\Delta_S(\mathcal{C})$ in this case. The definition of the SPO-RC loss can be naturally extended to the SPO-RC+ loss function as follows:

$$\ell_{\text{SPO-RC}_+}(\hat{\mathbf{c}}, \mathbf{c}; \hat{\mathcal{U}}, \mathbf{a}) := \text{cost}_+(\hat{\mathbf{c}}, \mathbf{c}; \hat{\mathcal{U}}) - \mathbf{c}^\top w^*(\mathbf{c}, \delta_{\mathbf{a}}).$$

Although the SPO-RC + loss function is continuous and convex, it serves only as a valid upper bound for the SPO-RC loss when $\mathbf{a} \in \hat{\mathcal{U}}$, as the SPO-RC loss is defined as $\Delta_S(\mathcal{C})$ when $\mathbf{a} \notin \hat{\mathcal{U}}$. Therefore, when using the SPO-RC+ loss for training, we must truncate the data to regions where $\mathbf{a} \in \hat{\mathcal{U}}$ to ensure that the loss function serves as a valid upper bound.

**Remark 2.5.** *Theorem 2.4 assumes that the hypothesis class $\mathcal{F}$ is well specified, which means $\mathbb{E}[\mathbf{c}|\mathbf{x}] \in \mathcal{F}$, which is rarely true in practice. Therefore, it can be advantageous to focus the predictive capacity of the model on regions where the feasibility of the solution is guaranteed (i.e., when $\mathbf{a} \in \hat{\mathcal{U}}$).*

# 3 SPO-RC+ with data truncation and importance reweighting

In this section, we introduce our main SPO-RC+ training procedure with data truncation to remove infeasible data points and importance reweighting to correct for the bias that this introduces.

## 3.1 Conformal prediction

We first introduce conformal prediction as a technique to construct the uncertainty set $\hat{\mathcal{U}}$ for the parameter $\mathbf{a}$ based on the context $\mathbf{x}$. Conformal prediction provides prediction sets that contain the true outcome with a specified probability $1 - \alpha$, independent of the data distribution and model assumptions. Specifically, we use the split conformal prediction approach, which begins by splitting the data into two sets: a training set $\mathcal{D}_{\mathbf{x},\mathbf{a}}^{n_p}$ and a calibration set $\mathcal{D}_{\mathbf{x},\mathbf{a}}^{n_c}$. The training set is used to fit a predictive model $\hat{g}$ that estimates $\mathbf{a}$ from $\mathbf{x}$. We then compute non-conformity scores in the calibration set to measure the discrepancy between predicted and actual values. For example, using the $\ell_2$ norm, the non-conformity score $s_i$ for a calibration point $(\mathbf{x}_i, \mathbf{a}_i)$ is calculated as $s_i = \|\mathbf{a}_i - \hat{g}(\mathbf{x}_i)\|_2$. These scores are used to determine a threshold $Q^{1-\alpha}$, the $\lceil \frac{(n_c+1)(1-\alpha)}{n_c} \rceil$ quantile of the set $\{s_i\}_{i \in \mathcal{D}_{\mathbf{x},\mathbf{a}}^{n_c}}$. This threshold defines the uncertainty set for new observations:

$$\mathcal{U}(\mathbf{x}) = \{\tilde{\mathbf{a}} : \|\tilde{\mathbf{a}} - \hat{g}(\mathbf{x})\|_2 \leq Q^{1-\alpha}\}.$$

The following proposition shows that the uncertainty set obtained via split conformal prediction achieves the desired marginal coverage.

**Proposition 3.1.** *Suppose the calibration dataset $\mathcal{D}_{\mathbf{x},\mathbf{a}}^{n_c}$ and the new observation $(\mathbf{x}_{\text{test}}, \mathbf{a}_{\text{test}})$ are i.i.d. samples from $\mathcal{D}_{\mathbf{x},\mathbf{a}}$ and $\mathcal{U}$ is constructed using split conformal prediction. Then, we have*

$$\mathbb{P}(\mathbf{a}_{\text{test}} \in \mathcal{U}(\mathbf{x}_{\text{test}})) \geq 1 - \alpha.$$

This result relies on the exchangeability of the calibration dataset and the new observation, which holds when they are i.i.d. samples. Importantly, Proposition 3.1 is true for any distribution $\mathcal{D}$ and any model $g$, regardless of its form or assumptions.

## 3.2 Importance reweighting

In Section 2.3, we introduced the SPO-RC and SPO-RC+ loss functions and suggested learning with truncated samples where $\mathbf{a} \in \hat{\mathcal{U}}$. However, the truncation causes a distribution shift, making it challenging to learn the true optimal solution $\mathbb{E}[\mathbf{c}|\mathbf{x}]$. Therefore, we use importance reweighting to correct the truncated distribution $\tilde{\mathcal{D}}$ to the true distribution $\mathcal{D}$ by reweighting the data points.

Suppose $\mathcal{U}$ is constructed using the split conformal prediction method. We consider learning from the truncated distribution $\tilde{\mathcal{D}}$, which satisfies $\mathbb{P}_{\tilde{\mathcal{D}}}(\mathbf{x}, \mathbf{a}, \mathbf{c}) \propto \mathbb{P}_{\mathcal{D}}(\mathbf{x}, \mathbf{a}, \mathbf{c})\mathbb{1}(\mathbf{a} \in \mathcal{U}(\mathbf{x}))$, where $\mathbb{1}(\cdot)$ is the indicator function. Notice that the minimizers $f^*_{\text{cost}}$ and $f^*_{\text{cost}_+}$ in (3) remain unchanged even when replacing the cost/cost$_+$ metrics with the SPO-RC/SPO-RC+ loss functions. Therefore, we choose to use the minimizers derived from the cost metrics, allowing us to focus on the marginal distributions $\tilde{\mathcal{D}}_{\mathbf{x},\mathbf{c}}$ and $\mathcal{D}_{\mathbf{x},\mathbf{c}}$. The difference between these marginal distributions can be decomposed into two components: (1) the difference in $\mathbb{P}_{\mathcal{D}}(\mathbf{x})$ versus $\mathbb{P}_{\tilde{\mathcal{D}}}(\mathbf{x})$, and (2) the difference in $\mathbb{P}_{\mathcal{D}}(\mathbf{c}|\mathbf{x})$ versus $\mathbb{P}_{\tilde{\mathcal{D}}}(\mathbf{c}|\mathbf{x})$. However, under the assumption that $\mathbf{c}$ and $\mathbf{a}$ are conditionally independent given $\mathbf{x}$, we find that $\mathbb{P}_{\mathcal{D}}(\mathbf{c}|\mathbf{x})$ and $\mathbb{P}_{\tilde{\mathcal{D}}}(\mathbf{c}|\mathbf{x})$ are the same.

**Assumption 3.2.** *We assume that conditioned on* $\mathbf{x}$*,* $\mathbf{c}$ *and* $\mathbf{a}$ *are independent. That is,*

$$\mathbb{P}_{\mathcal{D}}(\mathbf{c}, \mathbf{a}|\mathbf{x}) = \mathbb{P}_{\mathcal{D}}(\mathbf{c}|\mathbf{x})\mathbb{P}_{\mathcal{D}}(\mathbf{a}|\mathbf{x}) \ \forall \mathbf{x} \in \mathcal{X}.$$

**Lemma 3.3.** *Under Assumption 3.2, the conditional distributions of* $\mathbf{c}$ *given* $\mathbf{x}$ *in the original distribution* $\mathcal{D}$ *and the truncated distribution* $\tilde{\mathcal{D}}$ *are the same. That is,*

$$\mathbb{P}_{\mathcal{D}}(\mathbf{c}|\mathbf{x}) = \mathbb{P}_{\tilde{\mathcal{D}}}(\mathbf{c}|\mathbf{x}), \quad \forall \mathbf{x} \in \mathcal{X}.$$

Lemma 3.3 relies on the fact that the truncation leading to $\tilde{\mathcal{D}}$ depends only on $\mathbf{x}$ and $\mathbf{a}$, and not directly on $\mathbf{c}$. Consequently, the only relevant aspect is the marginal distribution of $\mathbf{x}$ in $\mathcal{D}$ and $\tilde{\mathcal{D}}$, reducing the problem to a simplified covariate shift problem.

$$\mathbb{E}_{(\mathbf{x},\mathbf{c})\sim\mathcal{D}_{\mathbf{x},\mathbf{c}}} \left[\text{cost}\left(f(\mathbf{x}), \mathbf{c}; \mathcal{U}(\mathbf{x})\right)\right] = \mathbb{E}_{(\mathbf{x},\mathbf{c})\sim\tilde{\mathcal{D}}_{\mathbf{x},\mathbf{c}}} \left[\frac{\mathbb{P}_{\mathcal{D}}(\mathbf{x}, \mathbf{c})}{\mathbb{P}_{\tilde{\mathcal{D}}}(\mathbf{x}, \mathbf{c})}\text{cost}\left(f(\mathbf{x}), \mathbf{c}; \mathcal{U}(\mathbf{x})\right)\right] =$$

$$\mathbb{E}_{(\mathbf{x},\mathbf{c})\sim\tilde{\mathcal{D}}_{\mathbf{x},\mathbf{c}}} \left[\frac{\mathbb{P}_{\mathcal{D}}(\mathbf{x})}{\mathbb{P}_{\tilde{\mathcal{D}}}(\mathbf{x})}\text{cost}\left(f(\mathbf{x}), \mathbf{c}; \mathcal{U}(\mathbf{x})\right)\right] = \mathbb{E}_{(\mathbf{x},\mathbf{c})\sim\tilde{\mathcal{D}}_{\mathbf{x},\mathbf{c}}} \left[\beta(\mathbf{x})\text{cost}\left(f(\mathbf{x}), \mathbf{c}; \mathcal{U}(\mathbf{x})\right)\right],$$

where $\beta(\mathbf{x}) = \frac{\mathbb{P}_{\mathcal{D}}(\mathbf{x})}{\mathbb{P}_{\tilde{\mathcal{D}}}(\mathbf{x})}$ is a importance weight. This relationship implies that, with proper importance weights, we can still learn the true optimal predictor $\mathbb{E}[\mathbf{c}|\mathbf{x}]$ despite the truncation. However, since the true underlying distributions $\mathcal{D}$ and $\tilde{\mathcal{D}}$ are unknown, we estimate $\beta$ using an empirical importance weighting method. While we use Kernel Mean Matching (KMM) (Huang et al., 2006) to estimate $\beta$, any other importance weight estimating methods such as logistic regression-based classifiers (Bickel et al., 2009) and adversarial learning approaches (Ganin et al., 2016) can be used. Given a truncated dataset $\tilde{\mathcal{D}}_{\mathbf{x},\mathbf{c}}^{n_s} = \{(\tilde{\mathbf{x}}_i, \tilde{\mathbf{c}}_i)\}_{i=1}^{n_s}$ and a target dataset $\mathcal{D}_{\mathbf{x},\mathbf{c}}^{n_t} = \{(\mathbf{x}_i, \mathbf{c}_i)\}_{i=1}^{n_t}$, KMM estimates the importance weight vector $\hat{\boldsymbol{\beta}} \in \mathbb{R}^{n_s}$ by solving:

$$\min_{\boldsymbol{\beta}} \frac{1}{2}\boldsymbol{\beta}^\top \mathbf{K} \boldsymbol{\beta} - \boldsymbol{\kappa}^\top \boldsymbol{\beta} \quad \text{s.t.} \quad \left|\sum_{i=1}^{n_s} \boldsymbol{\beta}_i - n_s\right| \leq n_s\epsilon, \ 0 \leq \boldsymbol{\beta}_i \leq B \ \forall i \in \{1, \cdots, n_s\}, \quad (5)$$

where $\mathbf{K} \in \mathbb{R}^{n_s \times n_s}$ is a kernel matrix with entries $\mathbf{K}_{ij} = k(\tilde{\mathbf{x}}_i, \tilde{\mathbf{x}}_j)$, $\boldsymbol{\kappa} \in \mathbb{R}^{n_s}$ is a vector with entries $\boldsymbol{\kappa}_i = \frac{n_s}{n_t}\sum_{j=1}^{n_t} k(\tilde{\mathbf{x}}_i, \mathbf{x}_j)$ and $k$ is a universal kernel. The parameters $B$ and $\epsilon$ control the range of the importance weights and the tolerance to match the distributions, respectively.

We present our learning algorithm within the SPO-RC framework in Algorithm 1. We use KMM to compute the importance reweighting vector $\hat{\beta}$ and the estimator $\hat{f}$ is then obtained by minimizing a weighted empirical cost$_+$ over the truncated data. To implement our approach, we partition the data into four distinct sets: two sets are allocated for split conformal prediction, while the other two are used for training and KMM. Notably, the dataset reserved for conformal prediction can also serve as the target dataset for KMM, thereby simplifying the data preparation process. Finally, we note that Appendix B includes generalization bounds that extend results of El Balghiti et al. (2023) to account for our context-dependent loss function structure and the fact that we may train on importance-reweighted truncated data.

**Algorithm 1:** SPO-RC+ with Data Truncation and Importance Reweighting

---

**Input:** $\hat{\mathcal{U}}, \mathcal{D}_1^{n_1} := \{(\mathbf{x}_i^1, \mathbf{c}_i^1, \mathbf{a}_i^1)\}_{i=1}^{n_1}, \mathcal{D}_2^{n_t} := \{(\mathbf{x}_i^2, \mathbf{c}_i^2, \mathbf{a}_i^2)\}_{i=1}^{n_t};$

Initialize $\mathcal{I} = \{\};$

**for** $i = 1$ **to** $n_1$ **do**

    **if** $\mathbf{a}_i^1 \in \hat{\mathcal{U}}(\mathbf{x}_i^1)$ **then**

        $\lfloor\ \mathcal{I} \leftarrow \mathcal{I} \cup \{i\};$

Define Source data set $\tilde{\mathcal{D}}_{\mathcal{I}} := \{(\mathbf{x}_i^1, \mathbf{c}_i^1, \mathbf{a}_i^1)\}_{i \in \mathcal{I}};$

$\hat{\boldsymbol{\beta}} \leftarrow$ Solve (5) with source dataset $\tilde{\mathcal{D}}_{\mathcal{I}}$ and target dataset $\mathcal{D}_2^{n_t};$

$\hat{f} \leftarrow \min\limits_{f \in \mathcal{F}} \sum\limits_{i \in \mathcal{I}} \frac{\hat{\boldsymbol{\beta}}(\mathbf{x}_i^1)}{|\mathcal{I}|} \mathrm{cost}_+\left(f(\mathbf{x}_i^1), \mathbf{c}_i^1; \hat{\mathcal{U}}(\mathbf{x}_i^1)\right);$

Return $\hat{f};$

---

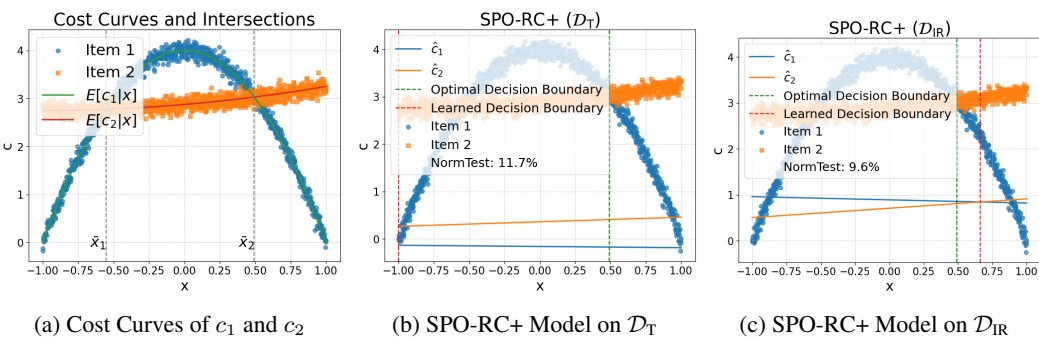

(a) Cost Curves of $c_1$ and $c_2$      (b) SPO-RC+ Model on $\mathcal{D}_{\mathrm{T}}$      (c) SPO-RC+ Model on $\mathcal{D}_{\mathrm{IR}}$

Figure 1: Visualization of the importance reweighting toy example

## 4 Numerical experiments

In this section, we present computational results from synthetic data on multiple applications. Unless otherwise specified, the $\ell_2$-norm is used as the conformal prediction score function in most experiments. Throughout, models are trained on three different datasets: the original dataset $\mathcal{D}_{\mathrm{O}}$, a truncated dataset $\mathcal{D}_{\mathrm{T}}$ where instances with $\mathbf{a}_i \notin \mathcal{U}(\mathbf{x}_i)$ are removed, and an importance-reweighted dataset $\mathcal{D}_{\mathrm{IR}}$ to adjust for the truncation bias. For KMM, we set the constant $B = 1000$ and $\epsilon = \frac{\sqrt{m}-1}{\sqrt{m}}$, where $m$ is the size of the truncated dataset. Although we choose a large value for $B$, in most cases the probability ratio did not exceed 20. A Gaussian kernel $k(\mathbf{x}_i, \mathbf{x}_j) = \exp\left(-\|\mathbf{x}_i - \mathbf{x}_j\|_2^2\right)$ is used. While KMM introduces some computational overhead, particularly for very large $m$, the number of samples in our experiments is medium to moderately large ($\approx 5000$ samples), and the primary computational cost comes from training the model under our proposed loss function, rather than from reweighting. The computation of KMM weights is relatively minor (about 1–4 seconds for $m = 1000$). Accordingly, our discussion on scalability primarily concerns training time with the SPO-RC+ loss.

Given a testing dataset $\mathcal{Z}^n = \{\mathbf{x}_i, \mathbf{c}_i, \mathbf{a}_i\}_{i=1}^n$, we assess model performance using the normalized metric:

$$\mathrm{NormSPORCTest}(\mathcal{Z}^n) := \frac{\sum_{i=1}^n \ell_{\mathrm{SPO-RC}}(\hat{\mathbf{c}}_i, \mathbf{c}_i; \mathcal{U}(\mathbf{x}_i), \mathbf{a}_i)}{\sum_{i=1}^n |\mathbf{c}_i^\top w^*(\mathbf{c}_i, \delta_{\mathbf{a}_i})|}.$$

If the induced solution $w^*(\hat{\mathbf{c}}_i, \mathcal{U}(\mathbf{x}_i))$ is infeasible, we set the loss $\ell_{\mathrm{SPO-RC}_+}(\hat{\mathbf{c}}_i, \mathbf{c}_i; \mathcal{U}(\mathbf{x}_i), \mathbf{a}_i)$ to the numerator value $|\mathbf{c}_i^\top w^*(\mathbf{c}_i, \delta_{\mathbf{a}_i})|$, thereby penalizing infeasibility. Our code is built on the open source software package PyEPO (Tang and Khalil, 2024).

**Toy examples.** To illustrate the effectiveness of importance reweighting, we present a simple toy example with two items. The value of each item is a function of the context $x$, as depicted in Figure 1a. Our goal is to predict which of the two items has the higher value using a linear regression model for each item. Suppose we randomly remove 30% of the data points where $|x| < 0.5$, simulating a

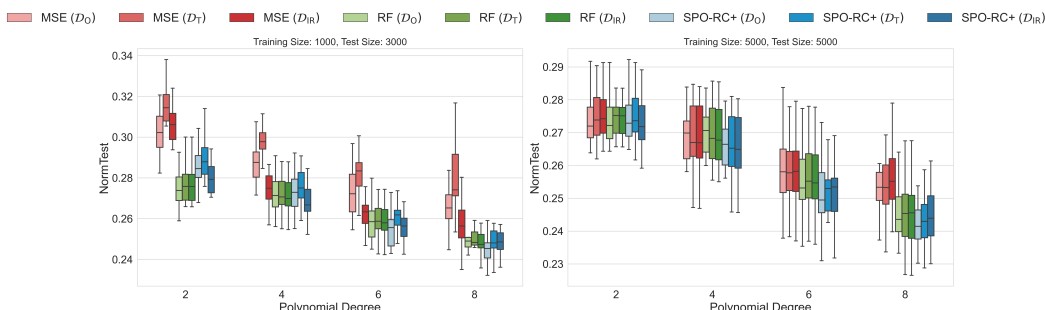

Figure 2: Out-of-sample test set (size 3000) NormSPORCTest values of linear models with MSE and SPO-RC+ loss functions, as well as random forests, on $\ell_2$ norm fractional knapsack instances.

Table 1: Percentage of infeasible solutions of fractional knapsack instances ($\ell_2$ norm) for different methods on a test set (size 3000), with $\deg_c = 4$.

| Models | PTO | MSE ($\mathcal{D}_{IR}$) | RF ($\mathcal{D}_{IR}$) | SPO-RC+ ($\mathcal{D}_{IR}$) |
|---|---|---|---|---|
| Infeasibility (%) | 45.00 | 0.02 | 0.02 | 0.02 |

truncation scenario. Figure 1b illustrates the resulting case where the linear models are trained using the truncated dataset $\mathcal{D}_T$. Due to truncation, the predicted value for item 1 consistently exceeds the predicted value for item 2 across all $x$, leading to incorrect selection of item 2 for certain values of $x$. This truncation effect, however, can be mitigated by training the models on an importance reweighted dataset $\mathcal{D}_{IR}$. As shown in Figure 1c, the learned decision boundary (indicated by the vertical red dotted line) closely aligns with the true optimal decision boundary (indicated by the vertical green dotted line), highlighting the effectiveness of the importance reweighting approach. Additional toy examples and detailed explanations can be found in Appendix C.1.

**Fractional knapsack problem using $\ell_2$ conformity score.** We consider the fractional knapsack problem, with robust formulation:

$$\max_{\mathbf{w} \in \mathbb{R}^d} \mathbf{c}^\top \mathbf{w} \quad \text{s.t.} \quad \mathbf{a}^\top \mathbf{w} \le b \, \forall \mathbf{a} \in \mathcal{U}(\mathbf{x}), \; \mathbb{1}_d^\top \mathbf{w} = 1, \; \mathbf{w} \in [0,1]^d,$$

where $\hat{\mathcal{U}}(x) = \{\tilde{\mathbf{a}} : \|\tilde{\mathbf{a}} - \hat{\mathbf{a}}\|_2 \le Q^{1-\alpha}\}$ is constructed using conformal prediction with $\ell_2$-norm, and $\hat{\mathbf{a}} = \hat{g}(\mathbf{x})$ is the estimated weight vector from the predictor $\hat{g}$. This problem can be reformulated as a second-order cone program (SOCP). We generate synthetic data with polynomial kernel models to describe the dependencies of $\mathbf{c}$ and $\mathbf{a}$ on $\mathbf{x}$; a detailed explanation of the data generation and model description can be found in Appendix C.2.

Figure 2 compares the out-of-sample NormSPORCTest performance of three different models, including linear models with either MSE or SPO-RC+ loss functions and random forest models, trained on three datasets $\mathcal{D}_O$, $\mathcal{D}_T$, and $\mathcal{D}_{IR}$, across varying levels of the polynomial degree dictating the complexity of the generated cost and constraint vectors. We observe that linear models trained with the SPO-RC+ loss have comparable performance to the non-parametric random forest model, and both outperform the linear regression model (MSE). (In this experiment only, we include random forests to illustrate the use of both parametric and nonparametric methods in our robust constraints setting. In later experiments, we focus on linear models only to better isolate and evaluate our core contributions.) In terms of dataset variations, models trained on $\mathcal{D}_{IR}$ yield better results in some cases, especially when the training size is small ($n = 1000$). Specifically, we observe that SPO-RC+ on $\mathcal{D}_{IR}$ outperforms SPO-RC+ on $\mathcal{D}_T$ in nearly every scenario. Table 1 presents the percentage of infeasible solutions for each model. The PTO model refers to the predict-then-optimize method, which uses the predicted $\hat{\mathbf{a}}$ directly without an uncertainty set. Table 1 shows that all methods that solve the robust constraint problem yield almost no infeasible solutions, while almost half of the solutions from the PTO approach are infeasible.

**Fractional knapsack problem using $\ell_1$ conformity score.** We consider the fractional knapsack problem with $\hat{\mathcal{U}}(x)$ constructed using conformal prediction with $\ell_1$-norm. The resulting robust

counterpart, constructed using the estimated $\hat{\mathbf{a}} = \hat{g}(\mathbf{x})$ and $\hat{\mathcal{U}}(x) = \{\tilde{\mathbf{a}} : \|\tilde{\mathbf{a}} - \hat{\mathbf{a}}\|_1 \leq Q^{1-\alpha}\}$, can be reformulated as a linear program (LP). Figure 3a compares the out-of-sample NormSPORCTest performance of different linear models using either MSE or SPO-RC+ loss functions and across the datasets $\mathcal{D}_O$, $\mathcal{D}_T$, and $\mathcal{D}_{IR}$. We observe that for small values of the polynomial degree, the parameter dictating the complexity of the generated cost and constraint vectors, the models trained with MSE slightly outperform those trained with SPO-RC+. However, the performance differences remain marginal in this regime. As expected by observations in the literature (Elmachtoub and Grigas, 2022), as the complexity of the relationship between features and cost vectors increases, the SPO-RC+ models consistently outperform the MSE models.

We also use this example to demonstrate how our methodology can be effectively combined with techniques to enhance scalability. The main scalability challenge of our method arises primarily during model training, particularly when using the SPO-RC+ loss function. As shown in Table 2, which reports the training times of our method with both MSE and SPO-RC+ losses on small and large datasets, the scalability of our approach is closely tied to the choice of the surrogate training loss employed. Consequently, our method directly benefits from any scalability improvements developed for decision-focused learning (DFL) methods. To demonstrate this, we adopt a randomized solution caching strategy inspired by Mulamba et al. (2021), who proposed caching solutions to reduce the number of solver calls during training. Specifically, in our case, each iteration of training requires evaluating the SPO-RC+ loss (at least once) and calculating a corresponding gradient by solving a robust optimization problem. For each data point, which has its own uncertainty set and corresponding robust problem, we maintain a cache of past solutions and, at every iteration, whenever needed, we either return the best solution from the cache or re-solve the problem. As the fraction of time that we perform a fresh re-solve gets smaller, Figures 3c and 3d demonstrate that the quality of models produced by the SPO-RC+ loss, across all three datasets, only slightly degrades, while the overall training time dramatically improves.

Table 2: Training time comparison (in seconds) for models using MSE and SPO-RC+ on the knapsack and alloy production problem. Results are reported for two instance sizes ($n = 5$ and $n = 10$), each trained on datasets of 1000 and 5000 training samples.

| Method | MSE ($\mathcal{D}_{IR}$) | MSE ($\mathcal{D}_{IR}$) | SPO-RC+ ($\mathcal{D}_{IR}$) | SPO-RC+ ($\mathcal{D}_{IR}$) |
|---|---|---|---|---|
| **Training data** | 1000 | 5000 | 1000 | 5000 |
| Knapsack (n=5) | 6.86 | 32.90 | 30.76 | 157.96 |
| Knapsack (n=10) | 8.67 | 35.65 | 44.36 | 214.60 |
| Alloy production (n=5) | 5.73 | 24.01 | 54.81 | 280.58 |
| Alloy production (n=10) | 6.98 | 33.24 | 95.04 | 427.95 |

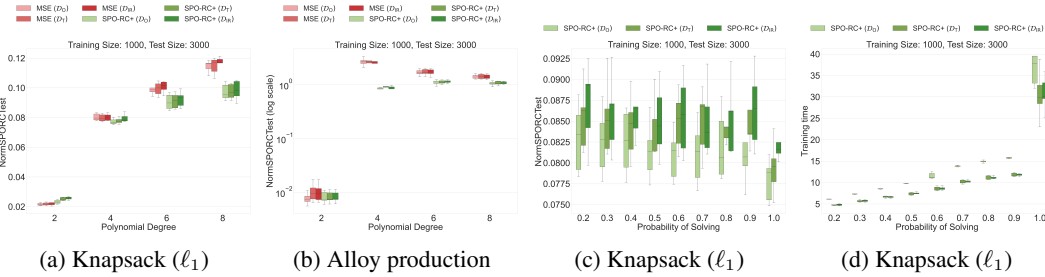

(a) Knapsack ($\ell_1$)  (b) Alloy production  (c) Knapsack ($\ell_1$)  (d) Knapsack ($\ell_1$)

Figure 3: Performance comparison across different tasks and evaluation criteria: (a)–(b) report out-of-sample test set (size 3000) NormSPORCTest performance under varying cost complexities for the fractional knapsack and alloy production problems, respectively. (c)–(d) evaluate the impact of the caching strategy in the knapsack problem with $\ell_1$-norm costs, using out-of-sample NormSPORCTest and training time as metrics.

**Alloy production problem (Hu et al. (2023)).**    In an alloy production process, a plant must produce a target alloy that requires a specified minimum quantity of different metals. Suppliers provide ores that contains a certain amount of the $m$ metals required for production. Since the true chemical

compositions of the ores are uncertain, we formulate the production problem as a robust covering LP:

$$\min_{\mathbf{w} \in \mathbb{R}^d} \mathbf{c}^\top \mathbf{w} \quad \text{s.t.} \quad \mathbf{a}_j^\top \mathbf{w} \geq h_j \, \forall \mathbf{a}_j \in \mathcal{U}_j(\mathbf{x}), \, j = 1, \cdots, m, \, \mathbf{w} \geq 0,$$

where $c_i$ is the cost per unit of ore from each supplier $i$, $h_j$ is the required quantity of metal $j$ and $a_j \in \mathbb{R}^m$ is the concentration vector of metal $j$, where each entry represents the concentration of metal $j$ in the ore across different suppliers. These concentrations may be uncertain but could be predicted based on certain chemical characteristics. We build an independent prediction model for each constraint, each using conformal prediction with $\ell_2$ score, leading to an SOCP reformulation of the robust problem above. A detailed explanation of the data generation process and model description can be found in Appendix C.3.

We run our experiments on a brass production scenario where two metals (Zinc and Copper) are required. Training times are reported in Table 2. The results in Figure 3b show a similar pattern as Figure 3a for the knapsack problem, where the MSE and SPO-RC+ models perform comparably for simpler cost functions, while SPO-RC+ consistently outperforms MSE as the complexity of the cost vector increases.

## 5 Conclusion and future directions

We propose the SPO-RC framework to learn the cost predictor in an integrated way while ensuring solution feasibility. The SPO-RC loss plays a central role by penalizing uncertainty sets $\hat{\mathcal{U}}$ that are either overly conservative or fail to capture the true parameters. While effective, the current approach has two key limitations: it applies only to problems with a linear objective, and it relies on a two-step procedure that learns the uncertainty set independently of the cost coefficients. Developing a related method to address nonlinear objectives and/or a fully joint learning procedure are promising directions for future work. Another potential direction for future work involves utilizing contextual uncertainty sets and robust constraints within other structured prediction applications, such as the work of Parmentier (2022) on learning simpler approximations of difficult variants of well-known optimization problems.

## Acknowledgments

This research was supported by NSF AI Institute for Advances in Optimization Award 2112533.

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

# A  Missing proofs

## A.1  Proof of Theorem 2.4

Our proof follows similar steps to those in Elmachtoub and Grigas (2022). Since the hypothesis class $\mathcal{F}$ is unrestricted, we can optimize the function values $f(\mathbf{x})$ individually for each $\mathbf{x} \in \mathcal{X}$. Therefore, solving the problems

$$f_{\mathrm{cost}}^* \in \arg\min_f \mathbb{E}_{(\mathbf{x},\mathbf{c}) \sim \mathcal{D}_{\mathbf{x},\mathbf{c}}} \left[ \mathrm{cost}\left( f(\mathbf{x}), \mathbf{c}; \mathcal{U}(\mathbf{x}) \right) \right],$$

$$f_{\mathrm{cost}_+}^* \in \arg\min_f \mathbb{E}_{(\mathbf{x},\mathbf{c}) \sim \mathcal{D}_{\mathbf{x},\mathbf{c}}} \left[ \mathrm{cost}_+\left( f(\mathbf{x}), \mathbf{c}; \mathcal{U}(\mathbf{x}) \right) \right],$$

is equivalent to optimizing each $f(\mathbf{x})$ separately. Consequently, for the remainder of the proof, we fix $\mathbf{x}$ to $\mathbf{x}_0$, and also $\hat{\mathcal{U}}_0 := \mathcal{U}(\mathbf{x}_0)$, and consider only the conditional distribution of $\mathbf{c}$. We define the risks associated with the cost and $\mathrm{cost}$ metrics as:

$$\begin{aligned}
R_{\mathrm{cost}}(\hat{\mathbf{c}}) &:= \mathbb{E}_{\mathbf{c}} \left[ \mathrm{cost}\left( \hat{\mathbf{c}}, \mathbf{c}; \hat{\mathcal{U}}_0 \right) \right], \\
R_{\mathrm{cost}_+}(\hat{\mathbf{c}}) &:= \mathbb{E}_{\mathbf{c}} \left[ \mathrm{cost}_+\left( \hat{\mathbf{c}}, \mathbf{c}; \hat{\mathcal{U}}_0 \right) \right],
\end{aligned} \tag{6}$$

where the $\mathbb{E}_{\mathbf{c}}$ denotes the expectation over $\mathbf{c}$. Let us define $\bar{\mathbf{c}} := \mathbb{E}_{\mathbf{c}}\left[\mathbf{c}|\mathbf{x}_0\right]$. We first list the propositions needed to complete the proof of Theorem 2.4.

**Proposition A.1** (Proposition 5 of Elmachtoub and Grigas (2022)). *If a cost vector $\mathbf{c}^*$ is a minimizer of $R_{cost}(\cdot)$, then $W^*(\mathbf{c}^*, \hat{\mathcal{U}}_0) \subseteq W^*(\bar{\mathbf{c}}, \hat{\mathcal{U}}_0)$. On the other hand, if $\mathbf{c}^*$ is a cost vector such that $W^*(\mathbf{c}^*, \hat{\mathcal{U}}_0)$ is a singleton and $W^*(\mathbf{c}^*, \hat{\mathcal{U}}_0) \subseteq W^*(\bar{\mathbf{c}}, \hat{\mathcal{U}}_0)$, then $\mathbf{c}^*$ is a minimizer of $R_{cost}(\cdot)$.*

**Proposition A.2** (Proposition 6 of Elmachtoub and Grigas (2022)). *Under Assumption 2.3, $\bar{\mathbf{c}}$ is the unique minimizer of $R_{cost_+}(\cdot)$.*

Since we have fixed $\mathbf{x}$ to $\mathbf{x}_0$, the uncertainty set $\hat{\mathcal{U}}_0$ is also fixed. Therefore, Propositions A.1 and A.2 reduce to those presented in Elmachtoub and Grigas (2022), and their proofs follow accordingly. Importantly, these propositions hold true when the constructed uncertainty set satisfies Assumption 2.3, regardless of whether the true parameter $\mathbf{a}$ lies within $\hat{\mathcal{U}}_0$ or not. This means we do not need to be concerned about the quality of $\hat{\mathcal{U}}_0$ to guarantee consistency when learning with the $\mathrm{cost}_+$ metric or SPO-RC+ loss function. (Indeed, recall the MSE loss will also guarantee consistency.) However, since we do not know the true distribution $\mathcal{D}$ and certain assumptions such as having a well-defined hypothesis class $\mathcal{F}$ often do not hold, this motivates us to focus on the region where feasibility is guaranteed and subsequently $\mathrm{cost}_+$ and SPO-RC+ yield valid upper bounds, as discussed in our main paper. We complete the proof of Theorem 2.4 using the above propositions.

*Proof.* Let $\mathbf{x}_0 \in \mathcal{X}$ be given and let $\hat{\mathcal{U}}_0 = \mathcal{U}(\mathbf{x}_0)$. By Proposition A.2, the expected cost vector $\mathbb{E}[\mathbf{c}|\mathbf{x}_0]$ is the unique minimizer of $R_{\mathrm{cost}_+}(\cdot)$. That is, $f_{\mathrm{cost}_+}^*(\mathbf{x}_0)$ is unique and $f_{\mathrm{cost}_+}^*(\mathbf{x}_0) = \mathbb{E}[\mathbf{c}|\mathbf{x}_0]$. Under Assumption 2.3, the optimal solution set $W^*(\mathbb{E}[\mathbf{c}|\mathbf{x}_0], \hat{\mathcal{U}}_0)$ is a singleton. Applying Proposition A.1, we conclude that $\mathbb{E}[\mathbf{c}|\mathbf{x}_0]$ is a minimizer of $R_{\mathrm{cost}}(\cdot)$. Since this holds for every $\mathbf{x} \in \mathcal{X}$, we have that almost surely $f_{\mathrm{cost}_+}^*$ is unique, $f_{\mathrm{cost}_+}^* = \mathbb{E}[\mathbf{c}|\mathbf{x}]$, and $f_{\mathrm{cost}_+}^*$ also minimizes $\mathbb{E}_{(\mathbf{x},\mathbf{c}) \sim \mathcal{D}_{\mathbf{x},\mathbf{c}}}\left[\mathrm{cost}\left(f(\mathbf{x}), \mathbf{c}; \mathcal{U}(\mathbf{x})\right)\right]$. This shows the Fisher consistency between the cost and $\mathrm{cost}_+$ metrics. Moreover, because the $f_{\mathrm{cost}}^*$ and $f_{\mathrm{cost}_+}^*$ remain the same when using SPO-RC and SPO-RC+ loss functions, respectively, this implies Fisher consistency between the SPO-RC and SPO-RC+ loss functions as well. $\square$

## A.2  Proof of Lemma 3.3

*Proof.* The truncated distribution $\tilde{\mathcal{D}}$ satisfies $\mathbb{P}_{\tilde{\mathcal{D}}}(\mathbf{x}, \mathbf{a}, \mathbf{c}) \propto \mathbb{P}_{\mathcal{D}}(\mathbf{x}, \mathbf{a}, \mathbf{c}) \mathbb{1}(\mathbf{a} \in \mathcal{U}(\mathbf{x}))$, where $\mathbb{1}(\cdot)$ is the indicator function. Therefore, we have

$$\begin{aligned}
\mathbb{P}_{\tilde{\mathcal{D}}}(\mathbf{x}, \mathbf{a}, \mathbf{c}) &\propto \mathbb{P}_{\mathcal{D}}(\mathbf{x}, \mathbf{a}, \mathbf{c}) \mathbb{1}(\mathbf{a} \in \mathcal{U}(\mathbf{x})) \\
&= \mathbb{P}_{\mathcal{D}}(\mathbf{c}|\mathbf{x}, \mathbf{a}) \mathbb{P}_{\mathcal{D}}(\mathbf{x}, \mathbf{a}) \mathbb{1}(\mathbf{a} \in \mathcal{U}(\mathbf{x})) \\
&= \mathbb{P}_{\mathcal{D}}(\mathbf{c}|\mathbf{x}, \mathbf{a}) \mathbb{P}_{\tilde{\mathcal{D}}}(\mathbf{x}, \mathbf{a}) \\
&= \mathbb{P}_{\mathcal{D}}(\mathbf{c}|\mathbf{x}) \mathbb{P}_{\tilde{\mathcal{D}}}(\mathbf{x}, \mathbf{a}) \text{ (by Assumption 3.2).}
\end{aligned}$$

To find the marginal distribution $\mathbb{P}_{\tilde{\mathcal{D}}}(\mathbf{x}, \mathbf{c})$, we sum over all possible $\mathbf{a}$:

$$\mathbb{P}_{\tilde{\mathcal{D}}}(\mathbf{x}, \mathbf{c}) = \mathbb{P}_{\tilde{\mathcal{D}}}(\mathbf{c}|\mathbf{x})\mathbb{P}_{\tilde{\mathcal{D}}}(\mathbf{x})$$
$$\propto \mathbb{P}_{\mathcal{D}}(\mathbf{c}|\mathbf{x})\mathbb{P}_{\tilde{\mathcal{D}}}(\mathbf{x}).$$

Dividing both sides by $\mathbb{P}_{\tilde{\mathcal{D}}}(\mathbf{x})$ gives $\mathbb{P}_{\tilde{\mathcal{D}}}(\mathbf{c}|\mathbf{x}) = \mathbb{P}_{\mathcal{D}}(\mathbf{c}|\mathbf{x})$.

$\square$

# B  Extended theoretical results

## B.1  Generalization bound

In this section, we present additional theoretical results, specifically generalization bounds for the cost metric. This extends the generalization bounds presented in El Balghiti et al. (2023) to accommodate context-dependent feasibility sets. We define the population risk of a function $f$ with respect to the cost metric as

$$\mathcal{R}_{\mathcal{D}}(f) := \mathbb{E}_{(\mathbf{x},\mathbf{c}) \sim \mathcal{D}_{\mathbf{x},\mathbf{c}}}\left[\text{cost}\left(f(\mathbf{x}), \mathbf{c}; \mathcal{U}(\mathbf{x})\right)\right],$$

and denote its empirical risk over $n$ samples collected in a dataset $\mathcal{D}^n$ as

$$\hat{\mathcal{R}}_{\mathcal{D}}^n(f) := \frac{1}{n}\sum_{i=1}^{n}\text{cost}\left(f(\mathbf{x}_i), \mathbf{c}_i; \mathcal{U}(\mathbf{x}_i)\right).$$

The multivariate Rademacher complexity $\mathfrak{R}_{\text{cost}}^n(\mathcal{F})$ (Bertsimas and Kallus, 2020) of the hypothesis class $\mathcal{F}$ with respect to the cost metric is defined as:

$$\mathfrak{R}_{\text{cost}}^n(\mathcal{F}) := \mathbb{E}_{\mathbf{x},\mathbf{c}}\mathbb{E}_{\sigma}\left[\sup_{f \in \mathcal{F}}\frac{1}{n}\sum_{i=1}^{n}\sigma_i\text{cost}\left(f(\mathbf{x}_i), \mathbf{c}_i; \mathcal{U}(\mathbf{x}_i)\right)\right],$$

where $\sigma_i$ are i.i.d Rademacher random variables for $i = 1, \cdots, n$. We denote the quantity $\Omega_S(\mathcal{C})$ as an upper bound on the maximum possible objective value over all solutions in $S$ across the entire space $\mathcal{C}$. Specifically, it is given by:

$$\Omega_S(\mathcal{C}) := \sup_{\mathbf{c} \in \mathcal{C}}\left(\max_{\mathbf{w} \in S}\mathbf{c}^\top\mathbf{w}\right).$$

By applying the cost metric to the renowned result from Bartlett and Mendelson (2002), we obtain the following theorem.

**Theorem B.1** (Theorem from (Bartlett and Mendelson, 2002)). *Let $\mathcal{F}$ be a hypothesis class and let $\delta > 0$. The following inequality holds with probability at least $1 - \delta$ over an i.i.d. dataset $\mathcal{D}^n$ for all $f \in \mathcal{F}$:*

$$\mathcal{R}_{\mathcal{D}}(f) \leq \hat{\mathcal{R}}_{\mathcal{D}}^n(f) + 2\mathfrak{R}_{\text{cost}}^n(\mathcal{F}) + \Omega_S(\mathcal{C})\sqrt{\frac{\log(1/\delta)}{2n}}. \tag{7}$$

In particular, when the hypothesis class $\mathcal{F}$ consist of linear functions, we can further bound the Rademacher complexity $\mathfrak{R}_{\text{cost}}^n(\mathcal{F})$ in terms of the sample size $n$ and other relevant quantities. We define $\mathbb{S} := \{\mathcal{S}(\mathbf{x}) : \mathbf{x} \in \mathcal{X}\}$ as the collection of all possible feasible sets and introduce the upper bound on their radius $\rho(\mathbb{S}) := \max_{\mathcal{S} \in \mathbb{S}}\max_{\mathbf{w} \in \mathcal{S}}\|\mathbf{w}\|_2$ to characterize the size of these sets.

**Proposition B.2** (Corollary 3 of El Balghiti et al. (2023)). *If $\mathcal{F}_{lin} := \{\mathbf{x} \to \mathbf{B}\mathbf{x}|\mathbf{B} \in \mathbb{R}^{d \times p}\}$ is the linear hypothesis class, then we have*

$$\mathfrak{R}_{\text{cost}}^n(\mathcal{F}_{lin}) \leq 2d\Omega_S(\mathcal{C})\sqrt{\frac{2p\log\left(2n\rho(\mathbb{S})d\right)}{n}} + O\left(\frac{1}{n}\right).$$

Notice that the extension can be easily made by adjusting the definition of $\rho(\mathbb{S})$, which characterizes the size of the feasible sets. By incorporating the Rademacher complexity bound from Proposition B.2 into (7), we obtain a generalization bound for the linear hypothesis class in our framework.

In addition to ensuring consistency when truncating, importance reweighting allows us to extend the generalization bounds presented in Theorem B.1. The following lemma provides a bound on the difference between the empirical risks calculated under the true distribution and the importance-reweighted truncated distribution.

Table 3: Comparison of NormSPORCTest and optimal decision boundary across different datasets

| Data | $R_A$ | $R_B$ | $R_C$ | Opt Boundary |
|------|-------|-------|-------|--------------|
| $\mathcal{D}_\mathrm{O}$ | **8.4%** | 12% | 11% | $x = \bar{x}_2$ |
| $\mathcal{D}_\mathrm{T}$ | 12.2% | **9.2%** | 15.8% | **Not Intersect** |
| $\mathcal{D}_\mathrm{IR}$ | **8.4%** | 12.2% | 11.2% | $x = \bar{x}_2$ |

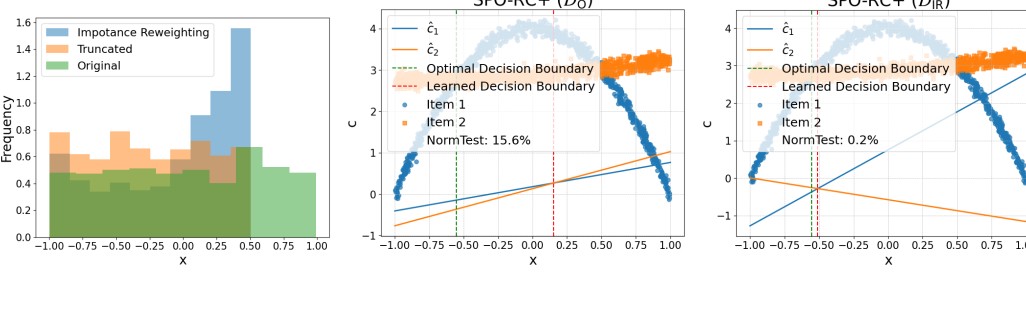

(a) Histograms      (b) SPO-RC+ Model on $\mathcal{D}_\mathrm{O}$      (c) SPO-RC+ Model on $\mathcal{D}_\mathrm{IR}$

Figure 4: Visualization of the truncation toy example

**Lemma B.3** (Lemma 4 from Huang et al. (2006)). *Suppose we have knowledge of $\beta(\mathbf{x}) \in [0, B]$ and let $\delta > 0$. With probability at least $1 - \delta$ over $n$ i.i.d. samples from the true distribution $\mathcal{D}$, and their corresponding truncations drawn from the truncated distribution $\tilde{\mathcal{D}}$, we have for all $f \in \mathcal{F}$:*

$$|\hat{\mathcal{R}}_\mathcal{D}^n(f) - \hat{\mathcal{R}}_{\beta\tilde{\mathcal{D}}}^n(f)| \leq (1 + \sqrt{2\log(2/\delta)})\Omega_S(\mathcal{C})\sqrt{\frac{B^2 + 1}{n}},$$

where $\beta\tilde{\mathcal{D}}$ represents the truncated distribution adjusted for the importance weight $\beta$. Using Lemma B.3, we can replace $\hat{\mathcal{R}}_\mathcal{D}^n(f)$ in the generalization bound (7) with $\hat{\mathcal{R}}_{\beta\tilde{\mathcal{D}}}^n(f)$, thus ensuring that our generalization analysis remains valid when using importance-reweighted truncated data.

**Proposition B.4.** *Suppose we have knowledge of $\beta(\mathbf{x}) \in [0, B]$, let $\mathcal{F}$ be a hypothesis class, and let $\delta > 0$. The following inequality holds with probability at least $1 - \delta$ over an i.i.d. dataset $\mathcal{D}^n$, and its corresponding truncated dataset $\tilde{\mathcal{D}}^n$, for all $f \in \mathcal{F}$:*

$$\mathcal{R}_\mathcal{D}(f) \leq \hat{\mathcal{R}}_{\beta\tilde{\mathcal{D}}}^n(f) + 2\mathfrak{R}_\mathrm{cost}^n(\mathcal{F}) + \Omega_S(\mathcal{C})\left(\sqrt{\frac{\log(1/\delta)}{2n}} + (1 + \sqrt{2\log(2/\delta)})\sqrt{\frac{B^2 + 1}{n}}\right).$$

## C    Additional experimental details and results

In this section, we provide additional details on the numerical experiments presented in Section 4 and some additional results. The experiments were conducted on a MacBook Pro equipped with an Intel chip and 16 GB of RAM.

### C.1    Additional details and results on toy examples

This section provides a more detailed explanation and additional toy examples supplementing the toy example presented in Section 4. To illustrate the effectiveness of importance reweighting and truncation, we consider a simplified version of the fractional knapsack problem. Specifically, we consider a case where our goal is to predict which of two items has a higher value based on a uniformly distributed context $x \in [-1, 1]$. The true relationships between the item values $c_1$ and $c_2$ and the context $x$ are given by:

$$c_1 = -4x^2 + 4 + \zeta_1, \ c_2 = \frac{1}{8}(x + 1)^2 + 2.75 + \zeta_2,$$

where $\zeta_1$ and $\zeta_2$ are i.i.d. normal random variables with mean 0 and standard deviation 0.1. We generate 1,000 samples of $(x, c_1, c_2)$. Figure 1a shows the scatter plot of these samples along with their true conditional expectations $\mathbb{E}[c_1|x]$ and $\mathbb{E}[c_2|x]$. The curves intersect at two points, denoted $\bar{x}_1$ and $\bar{x}_2$, partitioning the interval $[-1, 1]$ into three distinct regions: $A = [-1, \bar{x}_1]$, $B = [\bar{x}_1, \bar{x}_2]$, and $C = [\bar{x}_2, 1]$. Our goal is to use linear regression to predict $c_1$ and $c_2$ and make a decision based on which predicted value is higher. This is a special case of the fractional knapsack problem where the uncertain weight constraint is replaced with $w_1 + w_2 = 1$ without any uncertainty.

### C.1.1 Importance reweighting example

In regions $A$ and $C$, we observe that $c_2 < c_1$, whereas in region $B$, $c_1 < c_2$. Since the optimal decision changes across these regions and linear models may not capture the non-linear relationships perfectly, the decision made by the models will be incorrect in at least one of these regions. For instance, Figure 1c shows the learned decision boundary (red vertical line) of a linear model, which incorrectly predicts item 1 in region $A$, even though the true optimal choice is item 2. Notably, the optimal decision boundary (green vertical line in Figure 1c) makes incorrect decisions only in region $A$, where the difference between the two true curves is minimal. In fact, the area between the curves can be quantified using the NormSPORCTest metric on samples from each region, denoted as $R_A$, $R_B$, and $R_C$, corresponding to regions $A$, $B$, and $C$, respectively.

Suppose we randomly remove 30% of the data where $|x| < 0.5$, reducing samples in region $B$. This shifts the smallest region from $A$ to $B$ (as shown in Table 3) thereby changing the optimal decision boundary. However, by applying importance reweighting, we correct for this and realign the decision boundary with the true distribution, as shown in Table 3. Figure 1b and 1c illustrate the learned SPO-RC+ models on $\mathcal{D}_\mathrm{T}$ and $\mathcal{D}_\mathrm{IR}$. Notice that the learned decision boundary of SPO-RC+ on $\mathcal{D}_\mathrm{IR}$, the model with importance reweighting, is much closer to the true one than the model without it (SPO-RC+ on $\mathcal{D}_\mathrm{T}$). Additionally, the NormSPORCTest measured with 500 test samples was lower in SPO-RC+ on $\mathcal{D}_\mathrm{IR}$, indicating better performance.

### C.1.2 Truncation example

In this example, we compare SPO-RC+ on $\mathcal{D}_\mathrm{O}$ with SPO-RC+ on $\mathcal{D}_\mathrm{IR}$, to show the effectiveness of truncation. Building on the setting of the previous toy example, we introduce an additional capacity constraint $w_2 \le a_2$, where $a_2$ equals 100 if $x < 0.8$ and $a_2$ equals 0 otherwise. This makes the decision $(w_1, w_2) = (0, 1)$ infeasible when $x > 0.8$. We use a linear regression model to predict $a_2$ and construct the uncertainty set $\hat{\mathcal{U}}$ using conformal prediction with $\alpha = 0.25$. In the region where $x < 0.8$, we set $a_2$ large so that the constructed uncertainty set $\hat{\mathcal{U}}$ is large enough to include the original feasibility set defined by $w_1 + w_2 = 1$, for all $x \in [-1, 1]$. Thus, this example is equivalent to the previous one, except that the induced optimal solution can now be infeasible when $x > 0.8$.

Figure 4a shows the histograms of each dataset, indicating that all samples in region $C$ (including the possible infeasible range $[0.8, 1]$) are truncated. This means that for $x$ in the region $C$, $a_2 \notin \hat{\mathcal{U}}$. Figures 4b and 4c show the learned linear models from SPO-RC+ on $\mathcal{D}_\mathrm{O}$ and $\mathcal{D}_\mathrm{IR}$, respectively. The decision boundary learned from $\mathcal{D}_\mathrm{IR}$ closely matches the true optimal boundary, whereas the model trained on $\mathcal{D}_\mathrm{O}$ produces a less accurate solution. Evaluated on a dataset sampled from the feasibility-guaranteed region ($x < 0.5$), SPO-RC+ on $\mathcal{D}_\mathrm{O}$ performs poorly, achieving a NormSPORCTest of 16.7%. In contrast, SPO-RC+ on $\mathcal{D}_\mathrm{IR}$ performs almost perfectly, with a NormSPORCTest value of 0.2%. Furthermore, in the region $x > 0.8$, the induced solution of SPO-RC+ on $\mathcal{D}_\mathrm{O}$ is $(w_1^*, w_2^*) = (1, 0)$, which is infeasible. These results highlight the effectiveness of truncation when model complexity is limited and suggest focusing our learning capacity on the feasibility-guaranteed region.

### C.2 Additional details on the fractional knapsack instances

**Data generation process:** We consider a fractional knapsack problem with $d = 5$ items and $p = 10$ features. We adapt a popular data generation process in the literature, starting with Elmachtoub and Grigas (2022). We sample each element of the context vector $\mathbf{x} \in \mathbb{R}^p$ from the uniform distribution Uniform$(-1, 1)$, and generate the cost vector $\mathbf{c}$ as:

$$c_{ij} = \frac{5}{3.5^{\deg_c}} \left[ \left( \frac{(\mathbf{B}_c \mathbf{x}_i)_j}{\sqrt{p}} + 3 \right)^{\deg_c} + 10 \right] + \epsilon_{ij}^c,$$

where $i = 1, \ldots, n$ indexes the instances, $j = 1, \ldots, d$ indexes the items, and $\epsilon_{ij}^c \sim \mathcal{N}(0, 1)$. The matrix $\mathbf{B}_c \in {0, 1}^{d \times p}$ has elements sampled from a binomial distribution with probability $0.5$. The parameter $\deg_c$ is a fixed integer that determines the complexity of the cost function. Similarly, the weight vector $\mathbf{a}$ is generated as:

$$a_{ij} = \frac{5}{3.5^{\deg_a}} \left( \frac{(\mathbf{B}_a \mathbf{x}_i)_j}{\sqrt{p}} + 3 \right)^{\deg_a} + \frac{(p - \|\mathbf{x}_i\|_1)\epsilon_{ij}^a}{p},$$

where $\mathbf{B}_a$ and $\epsilon_{ij}^a$ are generated in the same manner as $\mathbf{B}_c$ and $\epsilon_{ij}^c$, respectively. Notice that the noise in $a_{ij}$ increases with $\|\mathbf{x}_i\|_1$. We fix $\deg_a = 4$ for the weight function, while varying $\deg_c = 2, 4, 6, 8$ to increase the complexity of the cost function, with $b = 20$.

**Model description for $\ell_2$-norm conformal prediction score:**  When the $\ell_2$-norm score is used for conformal prediction, the resulting robust reformulation takes the form of an SOCP:

$$\max_{\mathbf{w} \in \mathbb{R}^d} \mathbf{c}^\top \mathbf{w} \quad \text{s.t.} \quad Q^{1-\alpha} \|\mathbf{w}\|_2 \leq b - \hat{a}^\top \mathbf{w}, \ \mathbb{1}_d^\top \mathbf{w} = 1, \ \mathbf{w} \in [0, 1]^d.$$

To predict the weight vector $\hat{\mathbf{a}}$, we use a one-layer neural network with ReLU activation functions. We set $\alpha = 0.2$ to compute the conformal prediction quantile. For the cost vector $\hat{\mathbf{c}}$, we evaluate three models: Linear Regression (MSE), Random Forest (RF), and SPO-RC+ using a linear model. All models are trained using the Adam optimizer. The learning rate is set to $1 \times 10^{-3}$ for the MSE and RF models, and $4 \times 10^{-3}$ for the SPO-RC+ model. Training is conducted for 50 epochs with early stopping based on validation loss to prevent overfitting.

**Model description for $\ell_1$-norm conformal prediction score:**  When the $\ell_1$-norm score is used for conformal prediction, the resulting robust reformulation can be formulated as an LP:

$$\max_{\mathbf{w} \in \mathbb{R}^d} \mathbf{c}^\top \mathbf{w} \quad \text{s.t.} \quad Q^{1-\alpha} \|\mathbf{w}\|_\infty \leq b - \hat{a}^\top \mathbf{w}, \ \mathbb{1}_d^\top \mathbf{w} = 1, \ \mathbf{w} \in [0, 1]^d.$$

To predict the weight vector $\hat{\mathbf{a}}$, we use a one-layer neural network with ReLU activation functions, and use $\alpha = 0.2$ to compute the conformal prediction quantile. For the cost vector $\hat{\mathbf{c}}$, we evaluate both Linear Regression (MSE) and SPO-RC+ using a linear model. For the SPO-RC+ model, we warm start our method with MSE for faster training. All models are trained using the Adam optimizer. The learning rate is set to $1 \times 10^{-2}$ for the MSE and RF models, and $5 \times 10^{-2}$ for the SPO-RC+ model. Training is conducted for 50 epochs with early stopping based on validation loss to prevent overfitting.

### C.3  Additional details on the alloy production instances

**Data generation process:**  For the brass production scenario, we sample each element of the context vector and the cost vectors $\mathbf{c}$ using the same strategy used for the fractional knapsack problem. For each ore, we generate the concentration according to a Gamma distribution

$$G_{\cdot i} = \max \{\text{Gamma}(P_{\cdot i}) + \epsilon_k, 0\} \quad \text{for each supplier } i = 1, \ldots, d$$

where $P \in \mathbb{R}^{m \times d}$ is a base preference matrix independently sampled using a uniform distribution [0.1,1]. Since brass production requires around 30% of Zinc and 70% Copper to produce the alloy, we set the requirements to h = [2.9, 7.1].

**Model description:**  When the $\ell_2$-norm score is used for conformal prediction, the resulting robust reformulation can be formulated as an SOCP:

$$\min_{\mathbf{w} \in \mathbb{R}^d} \mathbf{c}^\top \mathbf{w} \quad \text{s.t.} \quad Q^{1-\alpha} \|\mathbf{w}\|_2 \leq \hat{a}_j^\top \mathbf{w} - h_j \ \forall j = 1, \cdots, m, \ \mathbf{w} \geq 0,$$

To predict the weight vector $\hat{\mathbf{a}}$, we use a one-layer neural network with ReLU activation functions. The conformal prediction quantile is computed using $\alpha = 0.2$. For the cost vector $\hat{\mathbf{c}}$, we evaluate two models: Linear Regression (MSE), and SPO-RC+, again using a linear model class and with the MSE solution as a warm start. All models are trained using the Adam optimizer. The learning

Table 4: Percentage of infeasible solutions of alloy production instances for different methods on a test set (size 3000), with $\deg_c = 8$.

| Models | PTO | MSE ($\mathcal{D}_{\mathbf{IR}}$) | SPO-RC+ ($\mathcal{D}_{\mathbf{IR}}$) |
|---|---|---|---|
| Infeasibility (%) | 30.894 | 0.194 | 0.188 |

rate is set to $1 \times 10^{-2}$ for MSE and $4 \times 10^{-2}$ for SPO-RC+. Training is run for 50 epochs with an adaptive learning rate scheduler and early stopping based on validation loss to prevent overfitting. Table 4 shows that all methods that solve the robustly constrained problem yield almost no infeasible solutions, while almost $30\%$ of the solutions from the PTO approach are infeasible.

