# OpenReview forum: "Smart Surrogate Losses for Contextual Stochastic Linear Optimization with Robust Constraints"
_NeurIPS.cc/2025/Conference — NeurIPS 2025 poster_

### Official Review · Reviewer_E3FA · 2025-07-01

**Clarity:** 3
**Significance:** 3
**Originality:** 3
**Rating:** 5
**Confidence:** 5

**Summary:**

The paper is motivated by providing a smart predict-then-optimize approach for chance-constrained contextual stochastic optimization:
$$\min_w\mathbb{E}(c^\top w|x) \text{ s.t. } \mathbb{P}(h(w,a) \leq 0|x) \geq 1-\alpha,$$
where $c$ and $a$ are random (and unobserved at decision time) but correlated with $x$.
A smart predict-then-optimize approach involves building a family of mappings (or conditional distributions) from context $x$ to decision $w$, and training it in an end-to-end manner, i.e., finding the mapping in the class that leads to the minimum expected cost with respect to the (unknown) distribution over $x$, $c$, and $a$.

The paper introduces a smart predict-then-optimize approach for contextual stochastic optimization with robust constraints:
$$\min_{w\in S}\mathbb{E}(c^\top w|x) \text{ s.t. } h(w,a) \leq 0, \text{ for all } a \in \mathcal{U}(x),$$
where $\mathcal{U}(x)$ is the robust feasibility set.
The robust feasibility set is defined (learned) a priori, using an estimate-then-optimize approach (as opposed to a smart predict-then-optimize approach). Here, a conformal prediction technique is used to build this uncertainty set.
Once this is fixed, a smart predict-then-optimize approach is used to learn the objective.
The SPO+ (structured hinge) loss is extended to this case.
Decisions to be imitated are obtained by solving the anticipative problem (where the random elements are known).
They may therefore not be feasible for the robust problem (since the constraints for the anticipative problem are weaker as they are not robust).
Infeasible solutions cannot be used as targets with the SPO+ loss.
An importance sampling-style reweighting of the feasible solutions in the training set is proposed to compensate for the fact that the infeasible ones have been removed.
Numerical experiments evaluate the results.

**Questions:**

- Why do you present the chance-constrained problem (1) first? Your methodology works only for the robust reformulation (2), which is generally not equivalent to (1) if we want the robust reformulation.
- How novel is the conformal approach for the robust set? The methodology looks standard.
- Line 92: "However, most of these approaches consider uncertainty only in the objective function rather than in the constraints." The challenge they represent for SPO approaches has been highlighted by [2]. They deal with a chance-constrained problem in the numerical experiments using an SPO approach that imitates the same single scenario solution. They note that chance constraints are poorly dealt with by using learning only for the objective (numerical experiments on chance-constrained CSP). Do you answer in some way to the challenge they raise? In other words, if we focus on the uncertainty in the constraint, does your SPO approach help in terms of the quality of the solution returned compared to using only the robust approach?
- Can you give examples of large-scale industrial applications where implementing your approach would be practical?
- Suppose you forget about the chance-constrained problem and take as input the (fully reformulated) robust problem. All the uncertainty is now in the objective. What is new in your SPO approach compared to applying directly ElMachtoub and Grigas' approach to that problem? Apart from the infeasible solutions... as feasible targets could perfectly be computed in that setting by taking the best solution (with respect to the cost known a posteriori) that is feasible for the robust problem.
- My first guess (and I would like you to explain why I am wrong) is that imitating the solution mentioned just above would give better performance than reweighting the training set only on the feasible solutions in the way you do. Indeed, it would use more of the information available. And the paper mentioned above underlined that SPO on the objective is bad at approximating the constraint, so using the realization of the uncertain parameter in the constraint is, in my opinion, not useful in practice.

**Ethical Concerns:**

["NO or VERY MINOR ethics concerns only"]

**Final Justification:**

This is a good paper. The contribution is more theoretical than practical.

**Limitations:**

- The main limitation, in my opinion, is that the uncertainty set $\mathcal{U}$ is not learned but given. As such, the resulting problem (for this fixed $\mathcal{U}$) can be recast as a usual contextual stochastic optimization problem. The only difference is that the decision we want to imitate may not be feasible (the reason why reweighting is needed) in the loss.
- Robust reformulations tend to be computationally complicated, so I guess the approach would not scale. Would it?
- Only linear objectives are addressed.

**Paper Formatting Concerns:**

No formatting concerns

**Quality:**

3

**Strengths And Weaknesses:**

### Strengths:
- Contextual stochastic optimization with uncertainty in the constraints is definitely interesting, especially the chance-constrained one (1), but also the second.
- The general methodology is sound, and the propositions make sense.
- While the different tools used are not new (SPO+ loss, importance sampling, conformal predictions), their combination for the robust problem is interesting.

### Weaknesses:
- The fact that the uncertainty set $\mathcal{U}$ is not learned is a key limitation in practice. Indeed, only the objective is learned in a Smart Predict-then-Optimize way, which is known to be a limitation. As a consequence, the method suffers from the same limits as robust approaches in terms of scalability (the problems are hard to solve practically).
- One could also argue that, because of this limitation, the improvement with respect to the literature is only incremental. Indeed, once $\mathcal{U}$ has been fixed, this is a classic SPO setting, and the paper only applies the classic methodology. The only difference is the fact that some anticipative solutions to imitate are not feasible, leading to the importance reweighting.
- The problems in the numerical experiments are small-scale.
- Some references are missing. In particular, the SPO+ loss was known 20 years before as the structured Hinge loss in the ML community [1, Chapter 6]. And Line 92, "However, most of these approaches consider uncertainty only in the objective function rather than in the constraints.". Such problems and the challenge they represent for SPO approaches have been highlighted by [2].

Since the difficulty in the chance-constrained optimization is linked to the constraints, and those are dealt with using a classic robust approach, the methodology is not truly a smart predict-then-optimize approach for the chance-constrained optimization problem, but rather for the objective of the problem with robust constraints. This does not prevent publication in my opinion, since the problem is already interesting, but it should be stated more clearly in Section 2 and the introduction (as it is already mentioned in the title).

[1] Nowozin, S., & Lampert, C. H. (2011). Structured learning and prediction in computer vision. Foundations and Trends® in Computer Graphics and Vision, 6(3–4), 185-365.

[2] Parmentier, A. (2022). Learning to approximate industrial problems by operations research classic problems. Operations Research, 70(1), 606-623.

---

> ### Author Rebuttal · Authors · 2025-07-31
>
> We first thank you for reading our paper and giving us valuable comments. Let us now answer your questions. Because of the word limit, we briefly quoted each weakness/question mentioned in the review.
>
> 1. **Since the difficulty in the chance-constrained optimization is linked to the constraints . $\cdots$ Why do you present the chance-constrained problem (1) first? $\cdots$**
>
> **A:** Thank you for this helpful observation. We agree with the reviewer’s interpretation of the problem we are addressing—that our methodology focuses on the objective optimization under robust constraints, rather than directly solving the chance-constrained optimization problem. We chose to present the chance-constrained problem (1) first to motivate the need for robustness and to provide a natural progression toward the robust formulation (2). Our goal was to illustrate how the robust approach emerges as a practical and tractable alternative when direct optimization under chance constraints is intractable or impractical. Due to space limitations, we were somewhat brief in clarifying this distinction in the current version. We will make sure to emphasize this point more clearly in the introduction and Section 2 of the camera-ready version.
>
> 2. **How novel is the conformal approach for the robust set? The methodology looks standard.**
>
> **A:** Thank you for your question. As discussed in the literature review, there has been growing interest within the operations research community in using conformal prediction to construct data-driven uncertainty sets. In our work, we adopt conformal prediction as one way to construct context-dependent uncertainty sets based on the available contextual features $x$, but we would like to emphasize that it is not the only method that can be used within our framework.
> To clarify the contribution of our paper: our goal is not to propose a novel method for constructing uncertainty sets per se, but rather to develop a unified framework that integrates existing techniques—conformal prediction and robust optimization—into our algorithm for decision-focused learning of the objective function. In this sense, we are leveraging well-established results from robust optimization and combining them with recent advances in predictive modeling to address a challenging class of contextual decision-making problems.
>
> 3. **Suppose you forget about the chance-constrained problem and take as input the (fully reformulated) robust problem. $\cdots$  My first guess (and I would like you to explain why I am wrong) is $\cdots$**
>
> **A:** Thank you for the detailed and thoughtful question.
> The key difference from the original SPO framework is that we model a context-dependent uncertainty set $\mathcal{U}(x)$, whereas the original SPO assumes a fixed feasible set across samples. This distinction is critical when constraint parameters are uncertain, because the feasible region varies with $x$.
> To the best of our understanding, we interpret the suggestion—“feasible targets could perfectly be computed in that setting by taking the best solution (with respect to the cost known a posteriori) that is feasible for the robust problem”—as corresponding to the approach of training on the full dataset without any truncation or reweighting. In our framework, this is exactly what we denote by $\mathcal{D}\_o$.
> In our experiments, we observe that the truncated and reweighted versions ($\mathcal{D}\_T$ and $\mathcal{D}\_{\text{IR}})$ do not consistently outperform $\mathcal{D}\_o$. This reinforces the point that truncation and reweighting are not always beneficial in practice-which aligns well with the reviewer’s opinion.
> Our primary motivation for introducing truncation and importance reweighting is theoretical: these steps allow us to derive generalization guarantees under sample selection bias. However, as our empirical results show, the practical gains are not guaranteed and may be affected by factors such as overfitting or poor estimation of the density ratio.
> In practice, we recommend selecting among $\mathcal{D}\_o$, $\mathcal{D}\_T$ and $\mathcal{D}\_{\text{IR}}$ based on validation performance, providing a flexible and pragmatic approach that adapts to the data at hand.
>
> 4. **Line 92: "However, most of these approaches consider uncertainty only in the objective function rather than in the constraints." The challenge they represent for SPO approaches has been highlighted by [2]. $\cdots$**
>
> **A:** Many thanks for pointing us to [2]. There are some interesting connections to that work that we had not considered before. Let us first comment on your question and then further on the connections to [2].
>
> In our view, our work takes the constraints (e.g., modeled with chance constraints or robust constraints) as hard constraints in the problem -- they must be satisfied with a very high probability. This justifies the robust approach, and leaves open the possibility of using a decision-focused (or SPO-type) approach for dealing with other parts of the problem, e.g. the objective function, that are risk-neutral. We demonstrate one way to accomplish this by forming the SPO-RC+ loss. In this way, our methodology ensures that the constraints will be satisfied with a high probability while the objective is learned in a decision-focused way. There is no guarantee of course that the chance constraints are dealt with in an ``optimal'' way as the robust approach is just a safe approximation and furthermore there is room in the future to also consider the possibility of learning the constraints in a decision-focused way (or adapting ideas from other works that do this), which we think is a valuable future direction.
>
> So to answer your question -- in a nutshell our approach is interesting and valuable when there is uncertainty in both the constraints and objective. Uncertainty in only the constraints reduces to robust optimization with a predictor that is not learned in a decision-focused way, and uncertainty only in the objective reduces to SPO. Our approach thus yields better quality solutions for the objective while ensuring feasibility w.r.t. the uncertain constraints with high probability.
>
> Based on our reading of [2], we think there is potential to apply an approach related to ours to the problem setting of that paper, although there are many open research questions clearly. To do so, two essential steps to apply our approach would be:  (i) replace imitating a single scenario solution with imitating a robust solution, e.g., using a robust problem as the simpler ``well-known'' problem, and (ii) one would have to consider a more involved cost structure in the objective, e.g., cost of delays or travel time, that is treatd in a risk-neutral way. One could possibly then use a standard model with conformal prediction to learn the constraints in a robust way while using a structured prediction type loss to learn the objective. Again, these are very high level ideas and many details remain, but this seems like a promising research direction to us.
>
> 5. **Some references are missing. In particular, the SPO+ loss was known 20 years before as the structured Hinge loss in the ML community [1, Chapter 6]**
>
> **A:** Thanks for pointing to these references, we appreciate it. We will certainly mention the connection to structured prediction, as pointed out in the original Elmachtoub and Grigas (2022) paper, in our camera ready version. The connection to [2] is quite interesting and, as mentioned above, is something that we had not considered before, so we appreciate that. We will certainly also mention this connection in our camera ready version.
>
> 6. **Can you give examples of large-scale industrial applications where implementing your approach would be practical?**
>
> **A:** While we have not (yet) collaborated with a real industrial partner to implement our algorithm, we feel that it is promising. Since our approach leverages well established methodologies -- including conformal prediction, robust optimization, and SPO+/structured hinge loss -- it can be smoothly integrated into existing pipelines. Furthermore, since this paper represents basic methodological research, our approach is applicable to a broad range of problems beyond those considered in the experiments, for example it could be applied to other optimization tasks such as battery storage operation [3]
>
> [3] Yeh, C., Christianson, N., Wu, A., Wierman, A., and Yue, Y. (2024). End-to-end conformal
> calibration for optimization under uncertainty. arXiv preprint arXiv:2409.20534.
>
> 7. **The main limitation, in my opinion, is that the uncertainty set $\mathcal{U}$ is not learned but given.**
>
> **A:** In our framework, the uncertainty set $\mathcal{U}(x)$ is learned once from data—by first training a constraint-parameter predictor $g(x)$ and applying conformal prediction to construct $\mathcal{U}(x)$—and is then held fixed while training the objective-cost model $f$; we believe the reviewer’s comment concerns this latter stage, and we view jointly learning $f, g,$ and an adaptive $\mathcal{U}$ that balances robustness and effectiveness as a promising direction for future work. In the paper’s conclusions and future-directions section, we acknowledge this limitation and flag this extension as a “fully joint learning procedure,” and we intend to pursue it by leveraging recent methods for data-driven uncertainty-set learning.
>
> 8. **Robust reformulations tend to be computationally complicated, so I guess the approach would not scale. Would it?**
>
> Thank you for your question. We agree with the reviewer that robust reformulations can introduce additional computational complexity. To mitigate this issue, we incorporate recent techniques in our implementation, such as warm-starting, which help reduce the overall computation time during training. These practical adjustments make our approach more scalable, especially in the context of the moderately sized problems considered in our experiments.

---

> > ### Comment · Reviewer_E3FA · 2025-08-06
> >
> > Thank you for these detailed responses that answer all my questions.

---

### Official Review · Reviewer_sXBi · 2025-07-02

**Clarity:** 4
**Significance:** 3
**Originality:** 3
**Rating:** 5
**Confidence:** 4

**Summary:**

This paper proposes Smart Predict-then-Optimize with Robust Constraints (SPO-RC), a framework that extends existing predict-then-optimize approaches to handle constraint uncertainty by constructing contextual uncertainty sets through conformal prediction.  In addition, the authors present a convex surrogate, SPO-RC$_+$, to SPO-RC and theoretically and empirically demonstrate the efficacy of their approach.

Overall, this paper's contribution is substantial, as the authors extend existing predict-then-optimize approaches to constraints in a sound and reasonable manner, which, to the best of my knowledge, has not been done within the contextual optimization literature to date.

**Questions:**

- Is there a reason within this particular setting that RF models would not be practically as scalable?
- Have the authors considered naive baselines of just penalizing constraint violation in the objective function and using standard SPO as a stronger baseline?
- Do the authors have comprehensive results on out-of-sample performance and feasibility for the alloy benchmark?
- Can the authors provide (and include in the appendix) the training times for all methods to give a clear picture of the scalability trade-offs?



### Minor Remarks
- In the camera-ready version, the appendix and main paper should be combined into a single document.
-  I have reviewed this paper at a previous conference, and the authors have made some progress towards addressing my concerns by including more experiments, although they are still limited.

**Ethical Concerns:**

["NO or VERY MINOR ethics concerns only"]

**Final Justification:**

See response to authors in comments below.

**Limitations:**

See weaknesses.

**Quality:**

3

**Strengths And Weaknesses:**

### Strengths
- **[S1]**: The methodological contributions of this paper are a significant strength.  Predict-and-optimize has become an increasingly growing area of research, and principled methods for problems with constraint uncertainty have remained elusive.  This paper bridges that gap by proposing one of the first approaches to addressing constraint uncertainty, building on SPO+ (a widely used loss function) through the establishment of tractable surrogates, training procedures, and appropriate theory.
- **[S2]**: Computational results.  Although the experiments have certain limitations, which will be discussed in the weaknesses section, the authors demonstrate the efficacy of their method on two benchmarks.
- **[S3]**: Clarity.  The paper is well-structured, and each aspect is well-motivated.


### Weaknesses
- **[W1]**: The computational experiments are relatively limited.  Both non-toy benchmarks are still extremely small instances, making any empirical claims, in my opinion, weak.  That said, this is a broader trend in the decision-focused learning literature, so this is within the standard, especially given that the authors consider constraint uncertainty.

- **[W2]**:  Scalability is another weakness of the suggested approach.  In contrast to existing predict-and-optimize approaches that solve a single-level linear or integer optimization problem, this method relies on solving a robust optimization problem, which is generally less efficient.  Given the authors claim that random forests not as scalable as linear models (technically accurate but not often not practically relevant) and avoid comparing to them on the alloy problem, I do encourage them to have a more transparent discussion around scalability limitations and transparent reporting of the time required to train each of the models.  It should be possible to mitigate these issues using approaches similar to those employed to improve efficiency in the objective uncertainty case. However, the paper would benefit from a discussion on this topic to inform readers about the limitations, rather than a single sentence in the contributions.

- **[W3]**:  Baselines/reporting.  The authors omit results for random forests for the alloy benchmark.  Additionally, essential metrics, such as infeasibility, are not presented for this benchmark.  I do acknowledge that the space is limited, but these could be given in the appendix for a more comprehensive evaluation.

The paper’s conceptual advance outweighs the empirical shortcomings, so I lean toward acceptance rather than rejection.

---

> ### Author Rebuttal · Authors · 2025-07-31
>
> We first thank you for reading our paper and giving us valuable comments. Let us now answer your questions.
>
> 1. **Is there a reason within this particular setting that RF models would not be practically as scalable?**
>
> **A:** Thank you for pointing this out. Our main reason for including Random Forest results in the knapsack example was to illustrate the use of both parametric and non-parametric methods in our robust constraints setting. Similar to Elmachtoub and Grigas (2022), we also wanted to showcase how the SPO-RC+ loss function can perform comparably to a non-parametric method like random forests, when there is enough training data. That being said, the more "apples to apples'' direct comparison is between the MSE and SPO-RC+ loss functions as they use the same architectures and will scale similarly modulo the challenges of dealing with the a more complicated loss function. We felt that the main point about random forests (and non-parameteric methods more broadly), which scales differently, was made on the first experiment we presented and therefore decided to drop this benchmark from additional experiments.
> We realize that the sentence in the current version of the paper —
> “(In this experiment only, we compare with random forests to demonstrate a comparison of linear models with a non-parametric method; as random forests is not as scalable, we drop it from the other experiments)” — is an oversimplification of the above discussion and may have caused confusion. In the revised version, we will clarify that scalability was not the main reason for excluding RFs from the later experiments. Rather, we wanted to keep the focus on models that best help evaluate the core contribution of our approach.
>
> 2. **Have the authors considered naive baselines of just penalizing constraint violation in the objective function and using standard SPO as a stronger baseline?**
>
> **A:** Thanks for the suggestion. We interpret your suggestion as referring to an exact or inexact penalty method, where the constrained optimization problem is approximately solved by incorporating the constraint into the objective function via a penalty term. This is an interesting suggestion, and something to possibly consider as a baseline, however we did not consider it due to its difficulties with ensuring feasibility when there is uncertainty in the constraint parameters. Indeed, if one expects a model to be able to perfectly predict $a$ based on $x$, i.e., there is no noise, then this approach may work well because it will be enough to ensure feasibility with respect to the predicted parameters. However, as soon as there is some uncertainty, i.e., some noise in the relationship, then other techniques are needed to ensure that the resulting decision will be feasible w.r.t. the true parameters $a$. This is a major motivation for the robust optimization methodology in the literature across many years and is our reasoning for considering our robust approach as well. In our paper, we primarily focus on scenarios where feasibility with a very high probability is a primary concern and thus we discarded many other approaches that do not promote a high probability of feasibility. For example, Table 1 shows that the PTO approach, which does something similar to what you suggest, has several orders of magnitude larger probability of infeasibility than our approaches. Note that we only considered the PTO approach in this one Table to illustrate that it does not promote feasibility and then discarded it as a baseline for this reason. We expect the penalty approach to be similar.
>
> 3. **Do the authors have comprehensive results on out-of-sample performance and feasibility for the alloy benchmark?**
>
> **A:** While we did not evaluate the RF model for the alloy production and knapsack benchmarks under an $\ell\_1$-based conformity score, most of the out-of-sample performance benchmarks are reported in Figure 3. As for the out-of-sample infeasibility results, they are consistent with those from the knapsack experiments under the $\ell\_2$-conformity score.
>
> | Dataset Metric       | PTO     | MSE ($\mathcal{D}\_\text{IR}$) | SPO-RC+ ($\mathcal{D}\_\text{IR}$) |
> |----------------------|---------|-------------------------------|-----------------------------------|
> | Infeasibility (%)    | 30.894  | 0.194                         | 0.188                             |
>
> **Table:** Percentage of infeasible solutions of alloy production instances for different methods on a test set (size 3000), with degc = 8.
>
>
> 4. **Can the authors provide (and include in the appendix) the training times for all methods to give a clear picture of the scalability trade-offs?**
>
> **A:** For the training times of the alloy production example, the following table summarizes the findings:
>
> | Method                  | MSE ($\mathcal{D}_\text{IR}$, 1000) | MSE ($\mathcal{D}_\text{IR}$, 5000) | SPO-RC+ ($\mathcal{D}_\text{IR}$, 1000) | SPO-RC+ ($\mathcal{D}_\text{IR}$, 5000) |
> |-------------------------|--------------------------------------|--------------------------------------|------------------------------------------|------------------------------------------|
> | Knapsack (n=5)          | 6.86                                 | 32.90                                | 30.76                                     | 157.96                                    |
> | Knapsack (n=10)         | 8.67                                 | 35.65                                | 44.36                                     | 214.60                                    |
> | Alloy production (n=5)  | 5.73                                 | 24.01                                | 54.81                                     | 280.58                                    |
> | Alloy production (n=10) | 6.98                                 | 33.24                                | 95.04                                     | 427.95                                    |
> **Table:** Training time comparison across methods and training data sizes.
>
> We will include the training times in the final version.

---

### Official Review · Reviewer_NQYV · 2025-07-03

**Clarity:** 2
**Significance:** 2
**Originality:** 2
**Rating:** 4
**Confidence:** 3

**Summary:**

Building on the SPO framework, the paper studies the SPO-RC loss and its convex surrogate to address CSLO problems with inequality constraints that depend on uncertain parameters. The paper establishes the Fisher consistency of the estimator and addresses bias caused by data truncation through importance reweighting techniques.

**Questions:**

Does the estimation error of the density ratio affect the coverage guarantee?

Could existing results on SPO be applied to approximately solve the problem by introducing an additional penalty term in the objective function?

**Ethical Concerns:**

["NO or VERY MINOR ethics concerns only"]

**Final Justification:**

Most concerns are addressed. One concern remains that the convex surrogate, the data truncation, and the subsequent importance-reweighting can degrade performance in finite samples, motivating the need for a more elegant solution. The manuscript is a valuable contribution to the community. The score is raised to 4.

**Limitations:**

The paper discusses limitations of the linear objective and the two-step procedure. The issue of sample inefficiency could merit further consideration.

**Quality:**

2

**Strengths And Weaknesses:**

The paper tackles the interesting problem of solving CSLO with inequality constraints that involve uncertain parameters. It proposes valid methodological components, including the use of conformal prediction for constructing uncertainty sets and importance reweighting for bias correction.

However, the paper lacks sufficient explanation of the SPO framework for readers unfamiliar with the concept. The approach may suffer from inefficiency, as the amount of data available for model training is reduced due to data truncation in SPO-RC+ and data splitting in conformal prediction.

---

> ### Author Rebuttal · Authors · 2025-07-31
>
> We first thank you for reading our paper and giving us valuable comments. Let us now answer your questions.
>
>
> 1. **However, the paper lacks sufficient explanation of the SPO framework for readers unfamiliar with the concept.**
>
> **A:** Thank you for this helpful suggestion. Due to space constraints, we were unable to provide a full explanation of the SPO framework in the current version, and we acknowledge that this may make the paper less self-contained for readers unfamiliar with the concept. As the areas of decision-focused learning and contextual stochastic optimization are maturing, we hope that many readers will be already be familiar with the framework and/or that our provided references will be helpful to gain the background knowledge.
>
>
> 2. **The approach may suffer from inefficiency, as the amount of data available for model training is reduced due to data truncation in SPO-RC+ and data splitting in conformal prediction.**
>
> **A:** Thank you for your insightful comment. We acknowledge that using split conformal prediction to construct the uncertainty set can reduce the effective size of the training dataset. However, we present split conformal prediction as one possible approach to build context-dependent uncertainty sets, chosen for its simplicity and well-established theoretical guarantees.
> We agree that exploring alternatives—such as full conformal prediction or other data-efficient methods—would be a valuable direction for future work to mitigate this issue.
> Additionally, we would like to clarify that the data truncation step in SPO-RC+ reduces the dataset by only a small constant fraction $\alpha$ in expectation (e.g., about $\alpha = 5\%$), which is relatively minor overall and compared to the reduction from data splitting in conformal prediction.
>
>
> 3. **Does the estimation error of the density ratio affect the coverage guarantee?**
>
> **A:** Thank you for the thoughtful question. We interpret the “coverage guarantee” in this context as referring to the convergence guarantee of our method—please feel free to correct us if this interpretation is inaccurate.
> Under this interpretation, we note that the estimation error of the density ratio does indeed affect the convergence guarantee of the importance-weighted variant of SPO-RC. Specifically, inaccurate estimation of the density ratio can lead to biased learning and the generalization bound guarantee might not hold.
> However, this does not impact the general convergence guarantees of the core SPO-RC method itself, which does not rely on importance reweighting.
>
> 4. **Could existing results on SPO be applied to approximately solve the problem by introducing an additional penalty term in the objective function?**
>
> **A:** We interpret your suggestion as referring to an exact or inexact penalty method, where the constrained optimization problem is approximately solved by incorporating the constraint into the objective function via a penalty term. This is an interesting suggestion, and something to possibly consider as a baseline, however we did not consider it due to its difficulties with ensuring feasibility when there is uncertainty in the constraint parameters. Indeed, if one expects a model to be able to perfectly predict $a$ based on $x$, i.e., there is no noise, then this approach may work well because it will be enough to ensure feasibility with respect to the predicted parameters. However, as soon as there is some uncertainty, i.e., some noise in the relationship, then other techniques are needed to ensure that the resulting decision will be feasible w.r.t. the true parameters $a$. This is a major motivation for the robust optimization methodology in the literature across many years and is our reasoning for considering our robust approach as well. In our paper, we primarily focus on scenarios where feasibility with a very high probability is a primary concern and thus we discarded many other approaches that do not promote a high probability of feasibility. For example, Table 1 shows that the PTO approach, which does something similar to what you suggest, has several orders of magnitude larger probability of infeasibility than our approaches.

---

> > ### Comment · Reviewer_NQYV · 2025-08-05
> >
> > Thanks for the addressing the concerns. The coverage guarantee in conformal prediction refers to the property that the constructed prediction sets contain the true response with at least the designated probability. Here it refers to the probability constraint in (1) and the reformulation in (2). It would be helpful if the author can briefly summarize how the convexification of the loss and the estimation error of the density ratio in reweighting affect the validity of this constraint under finite sample regime.

---

> > > ### Author Response · Authors · 2025-08-05
> > >
> > > Thank you for your comment and clarification. We want to clarify that our main algorithm, Algorithm 1, which is where the convexification of the loss function and the estimation error in the density ratio for reweighting come into play, actually assumes a *pre-processing step* using conformal prediction (or some other method for generating uncertainty sets). Thus, we assume that, going into Algorithm 1, there is a constraint parameter predictor $g$ and an uncertainty set model (learned with conformal prediction) that satisfies the coverage guarantee (Proposition 3.1). Therefore, the coverage guarantee will not be affected by the issues like convexification or estimation error for importance re-weighting. However, because Algorithm 1 involves truncation steps, importance reweighting, and the learning of the cost predictor $f$ via the SPO-RC$ ^+$ loss, the convergence guarantees for Algorithm 1 could conceivably be affected by these issues, which is what our original response referred to.

---

### Official Review · Reviewer_Gr2F · 2025-07-06

**Clarity:** 3
**Significance:** 4
**Originality:** 3
**Rating:** 5
**Confidence:** 4

**Summary:**

The paper introduces an extension to contextual stochastic linear optimization (CSLO) that incorporates uncertain constraints predicted by machine learning models. The authors propose a "Smart Predict-then-Optimize with Robust Constraints" (SPO-RC) approach, which uses contextual uncertainty sets to handle constraint uncertainty. This, in its turn extends SPO for problems with uncertainty in the constraints. This is done through the convex surrogate loss, SPO-RC+, and authors demonstrate its effectiveness through experiments on problems like fractional knapsack and alloy production. The approach includes data truncation and importance reweighting to address sample selection bias, showing improved performance in handling uncertainty.

**Questions:**

please see "Strengths And Weaknesses"

**Ethical Concerns:**

["NO or VERY MINOR ethics concerns only"]

**Final Justification:**

My concerns were addressed during rebuttal.

**Limitations:**

please see "Strengths And Weaknesses"

**Quality:**

3

**Strengths And Weaknesses:**

Overall, I think this is a nice addition to the literature of both domains (SPO and stochastic optimization), that potnetially can have a big impact...

Strengths

- The SPO-RC framework effectively incorporates uncertainty in constraints, which is a significant advancement over traditional methods that primarily focus on uncertainties in the objective function.

- The introduction of the SPO-RC+ loss as a convex surrogate allows for efficient optimization using first-order methods, making the approach computationally feasible for large-scale problems (similar to non-stochastic version of SPO+).

- The approach is validated through experiments on practical problems like fractional knapsack and alloy production, demonstrating its effectiveness in real-world applications. Although having large scale experiments would be nice.

Weaknesses

- The need for data truncation and importance reweighting adds layers of complexity to the implementation, which might be challenging for practitioners without a strong background in these techniques.

- My guess is that the use of importance reweighting to correct for sample selection bias could lead to overfitting, especially if the reweighting is not carefully calibrated.

- While the method is designed to be scalable, the additional computational overhead from constructing uncertainty sets and performing reweighting might limit its applicability to very large datasets or real-time decision-making scenarios. Would be great to validate this experimentally.

- The approach assumes certain properties of the data distribution (e.g., conditional independence of cost and constraint parameters), which may not hold in all practical situations, potentially affecting the robustness of the results.

---

> ### Author Rebuttal · Authors · 2025-07-31
>
> We first thank you for reading our paper and giving us valuable comments. Let us now answer your questions.
>
>
> 1. **The need for data truncation and importance reweighting adds layers of complexity to the implementation, which might be challenging for practitioners without a strong background in these techniques. My guess is that the use of importance reweighting to correct for sample selection bias could lead to overfitting, especially if the reweighting is not carefully calibrated.**
>
> **A:** We agree that data truncation and importance reweighting may seem to add complexity to the implementation. However, in practice, these steps can be implemented with only a few additional lines of code and do not significantly increase the computational burden for most standard learning pipelines.
> We would also like to clarify that our motivation for introducing truncation and importance reweighting is primarily theoretical: these components verify that our SPO-RC+ training loss is a valid upper bound on the true regret and they allow us to derive meaningful generalization bounds under sample selection bias. That said, as shown in our experimental results, the performance of the truncated and reweighted model (denoted as $\mathcal{D}\_{\text{IR}}$) does not always outperform its non-reweighted ($\mathcal{D}\_T$) and non-reweighted/non-truncated counterparts ($\mathcal{D}\_o$).
> This suggests that while the theoretical advantages are valid, practical performance may vary due to factors such as overfitting or inaccurate estimation of the density ratio used for reweighting -- issues that the reviewer rightly pointed out.
> In practice, one can choose the best among $\mathcal{D}\_o$, $\mathcal{D}\_T$, and $\mathcal{D}\_{\text{IR}}$ based on validation performance, which provides a pragmatic way to mitigate the potential drawbacks of reweighting or truncation when they are not beneficial.
>
> 2. **While the method is designed to be scalable, the additional computational overhead from constructing uncertainty sets and performing reweighting might limit its applicability to very large datasets or real-time decision-making scenarios. It would be great to validate this experimentally**
>
> **A:** Thank you for raising this important point. We agree that the reweighting step may introduce some computational overhead, particularly for very large datasets (e.g., with over 100,000 samples). However, in our experiments involving medium to moderately large datasets, we observed that the primary computational cost comes from training the model using our proposed objective, rather than from the reweighting step itself. The time required to compute the importance weights (generally taking around 1 to 4 seconds for 1000 samples) is relatively minor by comparison.
>
>
> 3. **The approach assumes certain properties of the data distribution (e.g., conditional independence of cost and constraint parameters), which may not hold in all practical situations, potentially affecting the robustness of the results.**
>
> **A:** Thank you for pointing out this important consideration. It is true that our method applies importance reweighting under the assumption of conditional independence between the cost and constraint parameters given the contextual variable $x$. However, this assumption is reasonable in settings where $x$ sufficiently captures the information that determines both $c$ and $a$—a common scenario in many practical applications where rich contextual features are available.
> We would also like to clarify that this assumption is only required for the theoretical justification of the importance reweighting step. It is not necessary for the convergence or validity of the core SPO-RC method itself. Even without importance reweighting, SPO-RC remains applicable.

---

### Decision · Program_Chairs · 2025-09-17

**Decision:**

Accept (poster)

**Comment:**

This paper introduces SPO-RC and its convex surrogate SPO-RC+, extending decision-focused learning to contextual stochastic optimization with uncertain constraints.
The contribution is regarded as significant and well-grounded: the theory is solid, the methodology integrates robust optimization with predict-then-optimize in a principled way, and the empirical validation, though limited in scale, was judged convincing. After rebuttal and discussion, all reviewers agreed that this is a strong paper and should be accepted.

For the camera-ready, the authors are encouraged to clarify connections to prior work (see reviews), expand the discussion of scalability and computational overhead, and provide additional details such as training times, infeasibility metrics. These improvements will increase the paper’s accessibility and impact.